# Backward Conformal Prediction

**Etienne Gauthier**[*]
INRIA-ENS-PSL Paris

**Francis Bach**
INRIA-ENS-PSL Paris

**Michael I. Jordan**
INRIA-ENS-PSL Paris
UC Berkeley

## Abstract

We introduce *Backward Conformal Prediction*, a method that guarantees conformal coverage while providing flexible control over the size of prediction sets. Unlike standard conformal prediction, which fixes the coverage level and allows the conformal set size to vary, our approach defines a rule that constrains how prediction set sizes behave based on the observed data, and adapts the coverage level accordingly. Our method builds on two key foundations: (i) recent results by Gauthier et al. [2025] on post-hoc validity using e-values, which ensure marginal coverage of the form $\mathbb{P}(Y_{\text{test}} \in \hat{C}_n^{\tilde{\alpha}}(X_{\text{test}})) \geq 1 - \mathbb{E}[\tilde{\alpha}]$ for any data-dependent miscoverage $\tilde{\alpha}$, and (ii) a novel leave-one-out estimator $\hat{\alpha}^{\text{LOO}}$ of the marginal miscoverage $\mathbb{E}[\tilde{\alpha}]$ based on the calibration set, ensuring that the theoretical guarantees remain computable in practice. This approach is particularly useful in applications where large prediction sets are impractical such as medical diagnosis. We provide theoretical results and empirical evidence supporting the validity of our method, demonstrating that it maintains computable coverage guarantees while ensuring interpretable, well-controlled prediction set sizes.

## 1   Introduction

Conformal prediction [Vovk et al., 2005] is a widely used framework for uncertainty quantification in machine learning. It produces set-valued predictions that are guaranteed to contain the true label with high probability, regardless of the underlying data distribution.

Given a calibration set of $n$ labeled examples $\{(X_i, Y_i)\}_{i=1}^n$ and a test point $(X_{\text{test}}, Y_{\text{test}})$, all assumed to be drawn from the same unknown distribution over $\mathcal{X} \times \mathcal{Y}$, conformal prediction constructs a prediction set $\hat{C}_n^\alpha(X_{\text{test}})$ such that

$$\mathbb{P}(Y_{\text{test}} \in \hat{C}_n^\alpha(X_{\text{test}})) \geq 1 - \alpha, \tag{1}$$

where the target miscoverage $\alpha \in (0, 1)$ is fixed by the practitioner. The probability is taken over both the calibration data and the test point, and the guarantee holds under the assumption that all $n+1$ points are exchangeable,[2] a generalization of the independent and identically distributed (i.i.d.) setting.

The method uses a score function $S : \mathcal{X} \times \mathcal{Y} \to \mathbb{R}_+$,[3] typically derived from a pre-trained model $f$, to evaluate how well the model's prediction $f(x)$ at input $x$ aligns with a candidate label $y$. In this paper, we assume that the scores are negatively oriented, meaning that a lower score indicates a better fit. The basic idea of conformal prediction is that, under exchangeability, the test score should behave like a typical calibration score: it should not stand out as unusually large. For a new input $X_{\text{test}}$,

---

[*]etienne.gauthier@inria.fr

[2]Exchangeable random variables are a sequence of random variables whose joint distribution is invariant under any permutation of their indices.

[3]In the conformal prediction literature, scores can also be negative. In this work, we explicitly assume nonnegativity to simplify the construction of e-variables like (3).

39th Conference on Neural Information Processing Systems (NeurIPS 2025).

the conformal set includes all labels $y$ such that the test score $S(X_{\text{test}}, y)$ is not excessively large compared to the scores from the calibration set. Standard methods rely on comparing score ranks, which can be interpreted in terms of p-values. We refer the reader to Angelopoulos and Bates [2023], Angelopoulos et al. [2024] for a recent overview of conformal prediction. Our work focuses on classification with a finite set of labels $\mathcal{Y}$, a setting that has been the focus of extensive research in conformal prediction [Lei, 2014, Sadinle et al., 2019, Hechtlinger et al., 2019, Romano et al., 2020, Angelopoulos et al., 2021, Cauchois et al., 2021, Podkopaev and Ramdas, 2021, Guan and Tibshirani, 2022, Kumar et al., 2023].

A key limitation of conformal prediction is that the size of the conformal set is entirely data-driven and *cannot be controlled in advance*. In many applications, this lack of control can be problematic: overly large prediction sets may be *too ambiguous to be useful*, especially in high-stakes or resource-constrained settings. This has motivated a growing body of work aiming to reduce the size of conformal sets while maintaining valid coverage guarantees [Stutz et al., 2022, Ghosh et al., 2023, Dhillon et al., 2024, Kiyani et al., 2024, Noorani et al., 2024, Yan et al., 2024, Braun et al., 2025].

In this work, we take a different perspective. Rather than fixing a desired coverage level $1 - \alpha$, we fix a *size constraint rule* $\mathcal{T}$ that flexibly determines, based on the observed calibration data $\{(X_i, Y_i)\}_{i=1}^n$ and the test feature $X_{\text{test}}$, the maximum allowable size of the prediction set. Formally:

**Definition 1.1** (Size constraint rule). A *size constraint rule* is a function

$$\mathcal{T} : (\{(X_i, Y_i)\}_{i=1}^n, X_{\text{test}}) \mapsto \mathcal{T}(\{(X_i, Y_i)\}_{i=1}^n, X_{\text{test}}) \in \{1, \ldots, |\mathcal{Y}|\},$$

where $|\mathcal{Y}|$ denotes the size of the label space. Given the observed calibration data $\{(X_i, Y_i)\}_{i=1}^n$ and test feature $X_{\text{test}}$, the rule $\mathcal{T}$ determines the maximum allowable size of the conformal set for $X_{\text{test}}$. An example of a size constraint rule is the constant size constraint rule, which corresponds to fixing a conformal set size independent from the observed data. In this case, $\mathcal{T}$ is a constant function, and by abuse of notation, we will also denote by $\mathcal{T} \in \{1, \ldots, |\mathcal{Y}|\}$ the constant value it returns.

The miscoverage level $\tilde{\alpha}$ is then chosen adaptively to satisfy this constraint. This gives rise to a new method, *Backward Conformal Prediction*, which ensures that the conformal set $\hat{C}_n^{\tilde{\alpha}}(X_{\text{test}})$ has size at most $\mathcal{T}(\{(X_i, Y_i)\}_{i=1}^n, X_{\text{test}})$, and provides marginal coverage guarantees of the form:

$$\mathbb{P}(Y_{\text{test}} \in \hat{C}_n^{\tilde{\alpha}}(X_{\text{test}})) \geq 1 - \mathbb{E}[\tilde{\alpha}], \tag{2}$$

where the probability is taken over both the calibration set and the test point. The guarantee above was established by Gauthier et al. [2025] by constructing conformal sets with e-values, a method known as conformal e-prediction [Vovk, 2025].

This paper then introduces a practical component: a novel *leave-one-out estimator* $\hat{\alpha}^{\text{LOO}}$ of the marginal miscoverage $\mathbb{E}[\tilde{\alpha}]$, computed from the calibration set. This allows practitioners to estimate the coverage guarantee in practice and make informed decisions about whether or not to trust the conformal set. The Backward Conformal Prediction procedure is illustrated in Figure 1.

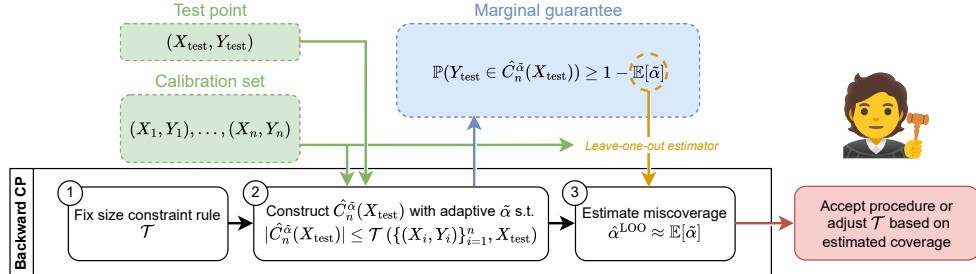

Figure 1: **Overview of Backward Conformal Prediction.** The procedure first fixes a (potentially data-dependent) size constraint rule $\mathcal{T}$, then constructs a conformal set $\hat{C}_n^{\tilde{\alpha}}(X_{\text{test}})$ using an adaptive miscoverage level $\tilde{\alpha}$ chosen to respect the size constraint. A leave-one-out estimator $\hat{\alpha}^{\text{LOO}}$ is computed on the calibration set to estimate the marginal miscoverage $\mathbb{E}[\tilde{\alpha}]$, enabling practitioners to decide whether to proceed with the current configuration or retune the size constraint $\mathcal{T}$ if the guarantee is unsatisfactory.

In traditional conformal prediction, the miscoverage level $\alpha$ is fixed in advance. Then, a conformal set satisfying marginal coverage guarantees (1) is constructed based on the calibration set. However, the size of the resulting conformal set is not controlled. In contrast, Backward Conformal Prediction reverses this workflow. It begins by defining a rule that dictates how the sizes of the conformal sets should behave, depending on the calibration data and the test feature. In this setting, the miscoverage level $\tilde{\alpha}$ becomes an adaptive quantity, dynamically adjusted based on the observed data, and is no longer fixed a priori. This adaptivity allows for controlling the size of the conformal set, though at the cost of indirect control over coverage. Nevertheless, the procedure still yields marginal coverage guarantees (2). This inversion of the standard conformal prediction design, where size control takes precedence over fixed coverage, motivates the name Backward Conformal Prediction.

Our approach is particularly useful in applications where prediction sets must be small and interpretable, such as medical diagnosis or inventory demand forecasting. We briefly present two motivating examples from these domains, where the control over prediction set sizes provides a natural and principled basis for decision-making.

**Healthcare.** In medical diagnosis, doctors must often infer a patient's condition $Y$ from their profile $X$ (symptoms, history, etc.) under time constraints. Standard conformal prediction, applied to calibration data $\{(X_i, Y_i)\}_{i=1}^n$, may produce prediction sets that are too large to have a practical impact on the diagnosis. Backward Conformal Prediction addresses this by allowing a size constraint which can be fixed or adaptive and data-dependent; for example, expanding the prediction set in rare or unusual cases while keeping it small and actionable for common cases. Our method ensures marginal coverage guarantees (2) and enables doctors to validate external reliability, via the guarantee that the coverage $1 - \hat{\alpha}^{\mathrm{LOO}} \approx 1 - \mathbb{E}[\tilde{\alpha}]$ remains above a desired level (e.g., 99%), striking a balance between diagnostic efficiency and rigor.

**Inventory demand forecasting.** In commerce, demand forecasting models predict daily sales $Y$ from features $X$ such as day of the week, weather, seasonality, and promotions. To forecast demand for a new day $X_{\mathrm{test}}$, a seller may wish to define a size constraint rule $\mathcal{T}$ that adapts to past demand variability in similar conditions: larger prediction sets for volatile periods (e.g., holidays) and smaller ones for stable days. Backward Conformal Prediction ensures valid coverage, and the seller can verify if estimated coverage $1 - \hat{\alpha}^{\mathrm{LOO}}$ meets reliability goals (e.g., 95%). This balances interpretability and reliability, supporting better stocking decisions under uncertainty.

## 2 Method

Backward Conformal Prediction builds on two core theoretical components. The first is the post hoc validity of the inverses of e-values, which provides marginal coverage guarantees while ensuring that the conformal sets have controlled size. The second is the estimation of these guarantees. While the first component has been established in conformal prediction by Gauthier et al. [2025], this paper establishes the second guarantee, which is essential for the method to be used in practice and not merely serve as a theoretical benchmark. Before explicating these two key steps, we first review the foundational principles of conformal prediction with e-values, or conformal e-prediction, which serves as the basis for Backward Conformal Prediction.

### 2.1 Conformal e-prediction

Most conformal prediction methods rely on the comparison of score ranks, which can be interpreted using p-values. However, it is also possible to construct conformal sets using e-values, a method known as conformal e-prediction [Vovk, 2025]. E-values are the values taken by e-variables:

**Definition 2.1** (E-variable). An *e-variable* E is a nonnegative random variable that satisfies

$$\mathbb{E}[E] \leq 1.$$

Backward Conformal Prediction works with any e-variable. For concreteness, we adopt the following e-variable, first introduced by Wang and Ramdas [2022] and Koning [2025b] and later used in the context of conformal prediction by Balinsky and Balinsky [2024], which is optimal in some sense [Koning, 2025b, Larsson et al., 2025]:

$$E^{\text{test}} = \frac{S(X_{\text{test}}, Y_{\text{test}})}{\frac{1}{n+1}\left(\sum_{i=1}^{n} S(X_i, Y_i) + S(X_{\text{test}}, Y_{\text{test}})\right)}. \tag{3}$$

It is straightforward to verify that $\mathbb{E}[E^{\text{test}}] = 1$ under exchangeability. The intuition, much like in standard conformal prediction methods, is that the test score $S(X_{\text{test}}, Y_{\text{test}})$ should not be excessively large compared to the calibration scores. However, rather than comparing ranks, as is typically done in conformal prediction, we directly compare the test score to the average of all the scores.

E-variables, when coupled with probabilistic inequalities, allow for the construction of conformal sets with valid $1 - \alpha$ coverage. For instance, by applying Markov's inequality, we get:

$$\mathbb{P}(E^{\text{test}} < 1/\alpha) = 1 - \mathbb{P}(E^{\text{test}} \geq 1/\alpha) \geq 1 - \alpha\mathbb{E}[E^{\text{test}}] \geq 1 - \alpha.$$

By carefully designing an e-variable based on the calibration set and the test point, we can construct useful conformal sets.

## 2.2 Post-hoc validity

E-values offer advantages that p-values alone cannot, including post-hoc validity, which enables more flexible and adaptive inference. The connections between e-variables and post-hoc validity have been explored, either explicitly or implicitly, in the following works: Wang and Ramdas [2022], Xu et al. [2024], Grünwald [2024], Ramdas and Wang [2024], Koning [2025a], and leveraged in conformal prediction by Gauthier et al. [2025]. For completeness, we briefly review the main application of post-hoc validity with e-variables in the context of conformal prediction.

**Proposition 2.2.** *Consider a calibration set $\{(X_i, Y_i)\}_{i=1}^{n}$ and a test data point $(X_{\text{test}}, Y_{\text{test}})$ such that $(X_1, Y_1), \ldots, (X_n, Y_n), (X_{\text{test}}, Y_{\text{test}})$ are exchangeable. Let $\tilde{\alpha} > 0$ be any miscoverage level that may depend on this data. Then, for the conformal set*

$$\hat{C}_n^{\tilde{\alpha}}(x) := \left\{ y : \frac{S(x, y)}{\frac{1}{n+1}\left(\sum_{i=1}^{n} S(X_i, Y_i) + S(x, y)\right)} < 1/\tilde{\alpha} \right\},$$

*the coverage inequality* (2) *holds, i.e.,*

$$\mathbb{P}(Y_{\text{test}} \in \hat{C}_n^{\tilde{\alpha}}(X_{\text{test}})) \geq 1 - \mathbb{E}[\tilde{\alpha}].$$

Note that when $\tilde{\alpha}$ is constant, we recover the standard conformal guarantee (1).

The flexibility of e-variables allows for the construction of conformal sets that adapt to the structure of the data, including explicit control over their size. In a classification task where we wish to control the conformal set sizes, standard conformal prediction techniques do not provide a direct way to enforce this. In contrast, using e-variables allows us to define a data-dependent miscoverage level $\tilde{\alpha}$, given a size constraint rule $\mathcal{T}$, as follows:

$$\tilde{\alpha} := \inf \left\{ \alpha \in (0, 1) : \# \left\{ y : \mathbf{E}_y^{\text{test}} < 1/\alpha \right\} \leq \mathcal{T}(\{(X_i, Y_i)\}_{i=1}^{n}, X_{\text{test}}) \right\}, \tag{4}$$

where, for the e-variable defined in Equation (3),

$$\mathbf{E}^{\text{test}} := \left( \frac{S(X_{\text{test}}, y)}{\frac{1}{n+1}\left(\sum_{i=1}^{n} S(X_i, Y_i) + S(X_{\text{test}}, y)\right)} \right)_{y \in \mathcal{Y}}$$

is the ratio vector for the test score, where each element corresponds to the ratio of the test score $S(X_{\text{test}}, y)$ to the average of the calibration scores and the test score. This formulation allows us to obtain coverage guarantees of the form (2) while explicitly controlling the size of the conformal set.

With additional assumptions on $\tilde{\alpha}$, such as sub-Gaussianity, it is possible to obtain coverage guarantees dependent on the observed miscoverage $\tilde{\alpha}$ and not on the marginal miscoverage $\mathbb{E}[\tilde{\alpha}]$ (see Boucheron et al. [2013] for a standard reference on concentration inequalities for sub-Gaussian variables). In this paper, we propose an approach that does not require any additional assumptions and relies solely on the estimation of the marginal miscoverage term $\mathbb{E}[\tilde{\alpha}]$. We suggest estimating it using the calibration data, which are fully accessible to the practitioner.

## 2.3 Leave-one-out estimator

In this section, we introduce the leave-one-out estimator $\hat{\alpha}^{\text{LOO}}$ for the marginal miscoverage term $\mathbb{E}[\tilde{\alpha}]$. Because it depends only on calibration scores and not on the test label, this estimator provides a fully computable, real-world proxy for miscoverage, enabling practitioners to obtain empirical guarantees on coverage. The key intuition is that, for large $n$, the denominator of the e-value $E^{\text{test}}$ in (3) closely approximates the expected score $\mathbb{E}[S(X, Y)]$, so $E^{\text{test}}$ effectively measures how far the test score deviates from the true average of the scores. To approximate the expectation of the marginal miscoverage, we emulate pseudo-ratios $\mathbf{E}^j$ based on the calibration set, where we compare $S(X_j, y)$ to the average of the calibration scores for all $y \in \mathcal{Y}$:

$$\mathbf{E}^j := \left( \frac{S(X_j, y)}{\frac{1}{n}\left(\sum_{i=1, i \neq j}^{n} S(X_i, Y_i) + S(X_j, y)\right)} \right)_{y \in \mathcal{Y}} \quad \text{for all } j = 1, \ldots, n.$$

We compute the corresponding pseudo-miscoverages:

$$\tilde{\alpha}_j := \inf \left\{ \alpha \in (0, 1) : \# \left\{ y : \mathbf{E}_y^j < 1/\alpha \right\} \leq \mathcal{T}(\{(X_i, Y_i)\}_{i=1, i \neq j}^{n}, X_j) \right\},$$

and we define:

$$\hat{\alpha}^{\text{LOO}} := \frac{1}{n} \sum_{j=1}^{n} \tilde{\alpha}_j. \tag{5}$$

This corresponds to averaging the $n$ pseudo-miscoverage terms obtained by artificially designating each calibration score $S(X_j, Y_j)$ as a pseudo-test score, with the remaining scores $\{S(X_i, Y_i)\}_{i=1, i \neq j}^{n}$ playing the role of the pseudo-calibration set. The construction of $\hat{\alpha}^{\text{LOO}}$ is schematically detailed in Figure 2.

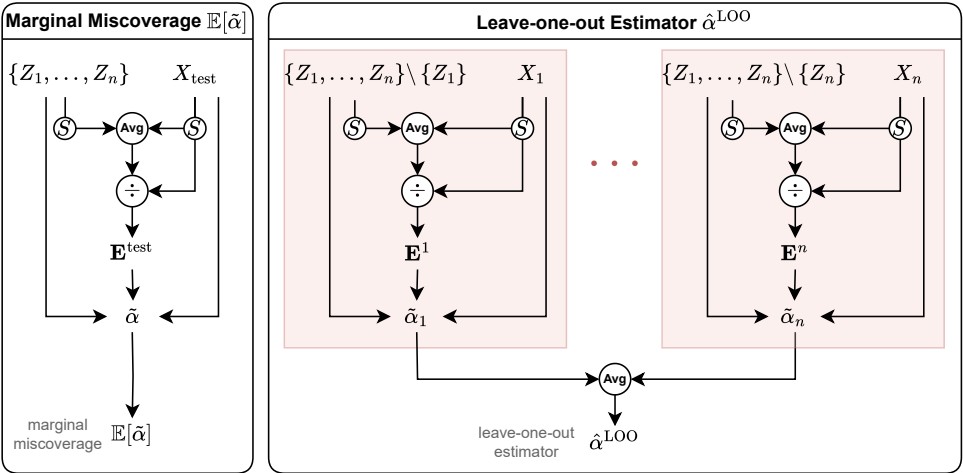

Figure 2: The **left panel** illustrates the definition of the true marginal miscoverage $\mathbb{E}[\tilde{\alpha}]$, which depends on the ratio between the test score $S(X_{\text{test}}, .)$ and the average of all $n+1$ scores. The **right panel** depicts the leave-one-out estimator $\hat{\alpha}^{\text{LOO}}$: each calibration point $j$ yields a pseudo-miscoverage $\tilde{\alpha}_j$ by comparing $S(X_j, .)$ to the average calibration score. Averaging these gives $\hat{\alpha}^{\text{LOO}}$, which approximates $\mathbb{E}[\tilde{\alpha}]$ without using the test score. Feature-label pairs are denoted $Z_i := (X_i, Y_i)$.

We summarize the Backward Conformal Prediction procedure in Algorithm 1.

---
**Algorithm 1:** Backward Conformal Prediction

---
**Input:** Calibration set $\{(X_i, Y_i)\}_{i=1}^n$, test feature $X_{\text{test}}$, size constraint rule $\mathcal{T}$, score function $S$
**Output:** Conformal set $\hat{C}_n^{\tilde{\alpha}}(X_{\text{test}})$ of size $\leq \mathcal{T}(\{(X_i, Y_i)\}_{i=1}^n, X_{\text{test}})$ and an approximate
   marginal coverage

**for** $i = 1$ **to** $n$ **do**
 $\lfloor$ Compute calibration score $S(X_i, Y_i)$;

Select $\tilde{\alpha}$ adaptively using Eq. (4) to build conformal set $\hat{C}_n^{\tilde{\alpha}}$ of size $\leq \mathcal{T}(\{(X_i, Y_i)\}_{i=1}^n, X_{\text{test}})$,
 satisfying guarantee (2);
Compute approximate miscoverage $\hat{\alpha}^{\text{LOO}}$ using Eq. (5);
**return** $\hat{C}_n^{\tilde{\alpha}}(X_{\text{test}})$, *approximate marginal coverage* $1 - \hat{\alpha}^{\text{LOO}}$

---

## 3 Theoretical analysis

In this section, we present theoretical properties satisfied by the estimator $\hat{\alpha}^{\text{LOO}}$ of the marginal miscoverage $\mathbb{E}[\tilde{\alpha}]$.

Let $\mu := \mathbb{E}[S(X, Y)]$ denote the expected value of $S(X, Y)$. For each calibration point $j \in \{1, \ldots, n\}$, we define the normalized score vector:

$$\tilde{\mathbf{E}}^j := \left( \frac{S(X_j, y)}{\mu} \right)_{y \in \mathcal{Y}},$$

which consists of the normalized score values of $S(X_j, y)$ for each $y \in \mathcal{Y}$. Similarly, for the test point, we define the vector:

$$\tilde{\mathbf{E}}^{\text{test}} := \left( \frac{S(X_{\text{test}}, y)}{\mu} \right)_{y \in \mathcal{Y}},$$

which represents the normalized score values $S(X_{\text{test}}, y)$ for each $y \in \mathcal{Y}$.

We show that, under certain assumptions on the size constraint rule $\mathcal{T}$, the estimator $\hat{\alpha}^{\text{LOO}}$ concentrates around its target $\mathbb{E}[\tilde{\alpha}]$, with an estimation error of order $O_P\left(\frac{1}{\sqrt{n}}\right)$ as the calibration size $n$ increases, using the $O_P$ notation from van der Vaart [1998].

### 3.1 Constant size constraint rule

To build intuition, we begin with the simple case where the size constraint rule $\mathcal{T}$ is a constant, which we also denote by $\mathcal{T}$ for convenience. This simplified case provides a foundation for understanding the estimator's properties under more general conditions.

**Theorem 3.1.** *Assume that the score function $S$ is bounded and takes values in the interval $[S_{\min}, S_{\max}]$, with $0 < S_{\min} \leq S_{\max}$. Suppose that the number of calibration samples satisfies $n > S_{\max}/S_{\min}$. In addition, assume that the vectors $\mathbf{E}^{\text{test}}$, $\mathbf{E}^j$, $\tilde{\mathbf{E}}^{\text{test}}$, and $\tilde{\mathbf{E}}^j$ satisfy the following properties almost surely:*

*(P1)* $1 \leq \# \{y \in \mathcal{Y} \mid \mathbf{E}_y < 1\} \leq \mathcal{T} < |\mathcal{Y}|$;

*(P2)* *For all $y \in \mathcal{Y}$, $\mathbf{E}_y \neq 1$.*

*Then, if the samples $(X_i, Y_i)$ are i.i.d., the leave-one-out estimator satisfies:*

$$\left| \hat{\alpha}^{\text{LOO}} - \mathbb{E}[\tilde{\alpha}] \right| = O_P\left(\frac{1}{\sqrt{n}}\right).$$

Note that, under the boundedness assumption on $S$, the vectors $\mathbf{E}^{\text{test}}$ and $\mathbf{E}^j$ are well defined. This assumption also implies $\mu \neq 0$, which guarantees that $\tilde{\mathbf{E}}^{\text{test}}$ and $\tilde{\mathbf{E}}^j$ are well defined.

In Property (P1), the inequality $1 \leq \# \{y \in \mathcal{Y} \mid \mathbf{E}_y < 1\}$ states that at least one label in the conformal prediction framework has a relatively low score. This ensures that the model does not assign high scores to all labels, which would make it difficult to differentiate between them. The assumption is

not overly restrictive, as it only requires one label to have a low score, leaving flexibility for other labels to have higher scores. The two other inequalities $\#\{y \in \mathcal{Y} \mid \mathbf{E}_y < 1\} \leq \mathcal{T} < |\mathcal{Y}|$ serve to rule out edge cases that would otherwise lead to degenerate behavior. Property (P2) plays a similar role. This assumption is mild, as score functions are typically continuous and attain any fixed value with probability zero. Likewise, the boundedness assumption on $S$ is generally unrestrictive; it helps ensure the stability of the analysis by excluding pathological situations.

The proof proceeds in several steps. First, we show that the infimum appearing in the definition of the miscoverage is stable under small perturbations of the ratio vectors $\mathbf{E}$. This is done by approximating the set cardinalities using a smooth surrogate based on bump functions, and applying the Implicit Function Theorem. Next, we show that the leave-one-out estimator $\hat{\alpha}^{\mathrm{LOO}}$ is close to the miscoverage $\mathbb{E}[\tilde{\alpha}]$, relying on the intuition that when $n$ is large, the average of the calibration scores is approximately equal to their expected value. We formalize this using Hoeffding's inequality to establish concentration (see Hoeffding [1963]). The full proof is deferred to Appendix C.

*Remark* 3.2. The explicit bound established in the proof of Theorem 3.1 shows that for any $\delta > 0$, we have

$$\left|\hat{\alpha}^{\mathrm{LOO}} - \mathbb{E}[\tilde{\alpha}]\right| \leq S_{\max} \frac{\sqrt{\frac{\log(4/\delta)}{2n}} + \frac{2}{n}}{\mu\left(S_{\min}/S_{\max} - \frac{1}{n}\right)} + \sqrt{\frac{\log(4/\delta)}{2n}} + \frac{2S_{\max}^2}{\mu S_{\min}} \frac{n+1}{n} \sqrt{\frac{\pi}{2(n+1)}},$$

with probability at least $1 - \delta$. While the constants $S_{\min}$ and $S_{\max}$ are typically known to the practitioner, $\mu$ is generally not observed. However, we may still upper bound the expression by

$$\left(\frac{S_{\max}}{S_{\min}}\right) \frac{\sqrt{\frac{\log(4/\delta)}{2n}} + \frac{2}{n}}{S_{\min}/S_{\max} - \frac{1}{n}} + \sqrt{\frac{\log(4/\delta)}{2n}} + 2\left(\frac{S_{\max}}{S_{\min}}\right)^2 \frac{n+1}{n} \sqrt{\frac{\pi}{2(n+1)}} =: R_\delta(n),$$

which guarantees that, as long as $\mathbb{P}(Y_{\mathrm{test}} \in \hat{C}_n^{\tilde{\alpha}}(X_{\mathrm{test}})) \geq 1 - \mathbb{E}[\tilde{\alpha}]$, we obtain

$$\mathbb{P}(Y_{\mathrm{test}} \in \hat{C}_n^{\tilde{\alpha}}(X_{\mathrm{test}})) \geq 1 - \hat{\alpha}^{\mathrm{LOO}} - R_\delta(n),$$

with probability at least $1 - \delta$. This provides a practical decision-making tool: given a target threshold $\tau$, the practitioner can trust the conformal set with probability at least $1 - \delta$ if the inequality $1 - \hat{\alpha}^{\mathrm{LOO}} - R_\delta(n) \geq \tau$ holds, and reject it otherwise.

In addition, the variance of $\hat{\alpha}^{\mathrm{LOO}}$ decreases at rate $O\left(\frac{1}{n}\right)$, further supporting its concentration around the marginal miscoverage $\mathbb{E}[\tilde{\alpha}]$.

**Theorem 3.3.** *Under the assumptions of Theorem 3.1, we have:*

$$\mathrm{Var}\left(\hat{\alpha}^{\mathrm{LOO}}\right) = O\left(\frac{1}{n}\right).$$

The proof of Theorem 3.3 can be found in Appendix C as well. The intuition is that the leave-one-out estimator $\hat{\alpha}^{\mathrm{LOO}}$ is a sum of terms, each of which is very close to independent terms, for which we know the variance of the sum. The remaining task is to bound the variance of the deviation, which we do by using a tail bound. This allows us to upper bound the variance of deviation through integration.

*Remark* 3.4. Note that when $\mathcal{T}$ is constant and the samples are i.i.d., coverage guarantees can be estimated in a straightforward way. For each calibration point $X_i$, let the corresponding prediction set consist of the top-$\mathcal{T}$ scores among $\{S(X_i, y)\}_{y \in \mathcal{Y}}$. One then checks whether the associated label $Y_i$ lies within this top-$\mathcal{T}$ set. The empirical frequency of miscoverage directly estimates the true miscoverage probability, and standard concentration inequalities can be used to bound its deviation. In contrast, the general case where $\mathcal{T}$ may depend on the calibration data itself requires a more sophisticated treatment. The simplified constant-size case provides an instructive foundation: it clarifies the main ideas behind our estimator and motivates the proof techniques developed in the more general, data-adaptive setting presented next.

## 3.2 General case

In the general case, we can also prove that the estimator $\hat{\alpha}^{\mathrm{LOO}}$ is consistent, provided that the size constraint rule preserves a form of stability for the miscoverages.

**Theorem 3.5.** *Assume that the score function $S$ is bounded and takes values in the interval $[S_{\min}, S_{\max}]$, with $0 < S_{\min} \le S_{\max}$. Suppose that the number of calibration samples satisfies $n > S_{\max}/S_{\min}$. In addition, assume that the size constraint rule and the score function satisfy the following stability property almost surely:*

*(P3) There exists a constant $L > 0$ such that:*

$$\left| \sum_{j=1}^n (\tilde{\alpha}_j - \mathbb{E}[\tilde{\alpha}]) \right| \le L \max_{y \in \mathcal{Y}} \left| \sum_{j=1}^n \left( \mathbf{E}_y^j - \mathbb{E}\left[ \tilde{\mathbf{E}}_y^{\text{test}} \right] \right) \right|.$$

*Then, if the samples $(X_i, Y_i)$ are i.i.d., the leave-one-out estimator satisfies:*

$$\left| \hat{\alpha}^{\text{LOO}} - \mathbb{E}[\tilde{\alpha}] \right| = O_P \left( \frac{1}{\sqrt{n}} \right).$$

Property (P3) implies that the size constraint rule preserves a form of stability between the variations in ratio vectors and their impact on miscoverage: small variations in the ratio vectors do not cause disproportionately large fluctuations in the miscoverages. Specifically, $\tilde{\alpha}_j$ depends on $\mathbf{E}^j$ and $\mathcal{T}(\{(X_i, Y_i)\}_{i=1}^n, X_{\text{test}})$, while $\tilde{\alpha}$ depends on $\mathbf{E}^{\text{test}}$ and $\mathcal{T}(\{(X_i, Y_i)\}_{i=1}^n, X_{\text{test}})$. When $\mathcal{T}$ is constant, as assumed in Section 3.1, we can obtain stability results for the miscoverages. However, when it is not constant, we require $\mathcal{T}$ to satisfy this stability condition to derive consistency results on miscoverages. This assumption was not required when $\mathcal{T}$ was constant, since the pseudo-miscoverages based on ratio vectors $\tilde{\mathbf{E}}^j$ we worked with were i.i.d., allowing us to directly apply concentration inequalities. In the general case, we rely on this stability assumption to directly manipulate the i.i.d. ratio vectors $\tilde{\mathbf{E}}^j$. Properties (P1) and (P2) are no longer needed, as they were only used to establish stability. The proof of Theorem 3.5 is given in Appendix C.

## 4   Experiments

We demonstrate through an image classification experiment how the estimator $\hat{\alpha}^{\text{LOO}}$ effectively approximates the true miscoverage $\mathbb{E}[\tilde{\alpha}]$, showcasing the effectiveness of Backward Conformal Prediction. In this section, we conduct experiments using a constant size constraint rule $\mathcal{T}$. Additional details and experiments are provided in Appendix B, starting with a binary classification example to motivate the need for controlling prediction set sizes, followed by an image classification experiment using a more complex data-dependent size constraint rule.

Our approach is evaluated on the CIFAR-10 dataset [Krizhevsky, 2009], which consists of 50,000 training images and 10,000 test images across 10 classes. For prediction, we use an EfficientNet-B0 model [Tan and Le, 2019] trained on the full training set. This model $f$ is treated as a black box that yields predictions. The model is trained to minimize the cross-entropy loss using stochastic gradient descent (SGD) with a learning rate of $0.1$, momentum $0.9$, weight decay $5 \times 10^{-4}$, and cosine annealing over 100 epochs. We use a batch size of 512 and apply standard data augmentation during training. At the end of training, the model $f$ achieves a training accuracy of $98.6\%$ and a test accuracy of $91.1\%$.

We evaluate the estimation $\hat{\alpha}^{\text{LOO}} \approx \mathbb{E}[\tilde{\alpha}]$ across various calibration sizes, $n \in \{100, 1000, 5000\}$, and prediction set sizes $\mathcal{T} \in \{1, 2, 3\}$. All the experiments are repeated $N = 200$ times. In each run, we sample a calibration set $\{(X_i, Y_i)\}_{i=1}^n$ of size $n$ uniformly at random from the test set. We then compute scores using the cross-entropy loss, defined by:

$$S(x, y) = -\log p_f(y|x),$$

where $p_f(y|x)$ is the predicted softmax probability by $f$ for class $y$ given feature $x$.

From the remaining test samples (i.e., not included in the calibration set), we draw one point $(X_{\text{test}}, Y_{\text{test}})$ uniformly at random to serve as the test point. Based on the test features and the calibration set, we construct a conformal set $\hat{C}_n^{\tilde{\alpha}}(X_{\text{test}})$ of size at most $\mathcal{T}$, along with an estimated coverage level $1 - \hat{\alpha}^{\text{LOO}}$, using the output of Algorithm 1. Both $\tilde{\alpha}$ and the intermediate $\tilde{\alpha}_j$ used to compute $\hat{\alpha}^{\text{LOO}}$ are computed via binary search over $\alpha \in (0, 1)$: we seek the smallest $\alpha$ such that the number of labels $y$ with $\mathbf{E}_y^{\text{test}} < 1/\alpha$ (respectively, $\mathbf{E}_y^j < 1/\alpha$) is at most $\mathcal{T}$. The search stops when the candidate value of $\alpha$ is within a tolerance of $0.005$ from the optimal value.

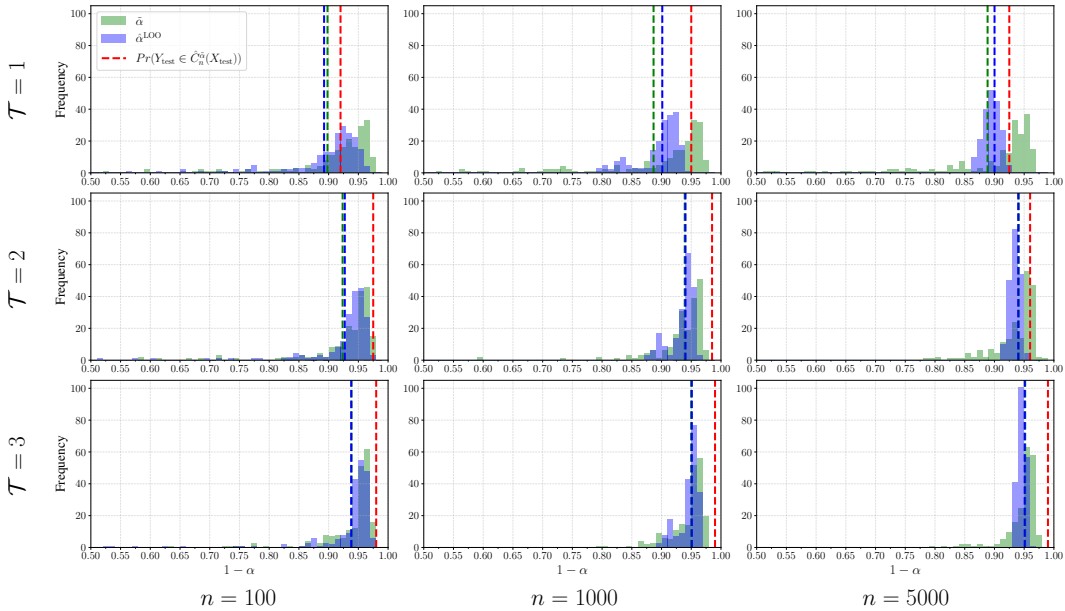

Figure 3: Histograms of $1 - \tilde{\alpha}$ and $1 - \hat{\alpha}^{\mathrm{LOO}}$ from $N = 200$ runs for various $(n, \mathcal{T})$ configurations. The red dashed line shows the empirical coverage probability. See text for details.

We report the results averaged over the $N$ runs in Figure 3. We plot the histogram of the $N$ values of $1 - \tilde{\alpha}$ obtained across runs in green, along with their expected value $1 - \mathbb{E}[\tilde{\alpha}]$ shown as a green dashed line. Overlaid on this, we display the histogram of the corresponding $N$ leave-one-out estimators $1 - \hat{\alpha}^{\mathrm{LOO}}$ in blue. A blue dashed vertical line indicates the average value of $1 - \hat{\alpha}^{\mathrm{LOO}}$ for visualization. We also plot the empirical coverage rate $\mathbb{P}(Y_{\mathrm{test}} \in \hat{C}_n^{\tilde{\alpha}}(X_{\mathrm{test}}))$ as a red dashed line.

Recall that the true coverage probability $\mathbb{P}(Y_{\mathrm{test}} \in \hat{C}_n^{\tilde{\alpha}}(X_{\mathrm{test}}))$ satisfies the coverage inequality (2), which is confirmed in our experiments: the empirical coverage probability, depicted by the red dashed line, consistently exceeds the empirical coverage $1 - \mathbb{E}[\tilde{\alpha}]$, shown by the green dashed line. Furthemore, while the values of $1 - \tilde{\alpha}$ can exceed the coverage probability, especially for small $n$, the coverage always remain above $1 - \mathbb{E}[\tilde{\alpha}]$, aligning with our theoretical marginal guarantees. We also observe that the leave-one-out estimators $\hat{\alpha}^{\mathrm{LOO}}$ closely approximate the marginal miscoverage $\mathbb{E}[\tilde{\alpha}]$, with the approximation improving as the calibration size $n$ increases, as suggested by Theorems 3.1 and 3.3. Note that the histograms of $\hat{\alpha}^{\mathrm{LOO}}$ and $\tilde{\alpha}$ may differ, but this is expected, as $\hat{\alpha}^{\mathrm{LOO}}$ targets $\mathbb{E}[\tilde{\alpha}]$ rather than higher-order properties. These results show that practitioners can effectively make use of available calibration data to approximate marginal coverage guarantees through the leave-one-out estimator.

## 5 Conclusion

We introduced Backward Conformal Prediction, a novel framework that prioritizes controlling the prediction set size over guarantees of fixed coverage levels. Built on e-value-based inference, our method enables data-dependent miscoverage and uses a leave-one-out estimator to approximate marginal miscoverage. This approach offers a flexible, principled alternative in settings where interpretability and set size control are critical, such as medical diagnosis. Our theoretical and empirical results show valid coverage guarantees while enforcing a user-specified size constraint. These findings open new avenues for adaptive conformal prediction and data-driven control that go beyond mere coverage.

Backward Conformal Prediction differs from standard conformal prediction in a fundamental way: it allows practitioners to control the size of prediction sets while simultaneously estimating coverage. In standard conformal prediction, coverage is guaranteed, but the size of each prediction set is uncontrolled and can vary widely. Our approach provides more actionable information, enabling practitioners to adjust the size constraint if the resulting coverage is too low. While the coverage

guarantee in Backward Conformal Prediction might be slightly conservative due to the use of Markov's inequality, this trade-off allows for principled decisions about the balance between set size and reliability, offering a more informative and flexible framework for practical applications.

While we focused on miscoverage definitions with fixed set sizes, the method is applicable to any data-dependent miscoverage. Future work could explore alternative formulations and investigate real-world applications under different constraint rules. Additionally, the stability assumption in Theorem 3.5 could likely be relaxed under regularity conditions.

The size constraint rule also affects the marginal miscoverage $\mathbb{E}[\tilde{\alpha}]$, and dynamic, anytime adjustments to this rule based on the test sample could be explored, leveraging the leave-one-out estimator for real-time miscoverage estimation.

Finally, although our analysis focuses on classification, the method can also be applied to regression, which could be a valuable direction to explore.

## Acknowledgments and Disclosure of Funding

The authors would like to thank the reviewers for their thoughtful comments.

Funded by the European Union (ERC-2022-SYG-OCEAN-101071601). Views and opinions expressed are however those of the author(s) only and do not necessarily reflect those of the European Union or the European Research Council Executive Agency. Neither the European Union nor the granting authority can be held responsible for them.

This publication is part of the Chair "Markets and Learning," supported by Air Liquide, BNP PARIBAS ASSET MANAGEMENT Europe, EDF, Orange and SNCF, sponsors of the Inria Foundation.

This work has also received support from the French government, managed by the National Research Agency, under the France 2030 program with the reference "PR[AI]RIE-PSAI" (ANR-23-IACL-0008).

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

# A  Further related work

In this paper, we focus on the split conformal prediction setting [Papadopoulos et al., 2002], where proper training data is used to pre-train a model, and a separate calibration set is then formed from other data to construct conformal sets. Conformal prediction has become a widely adopted technique due to its reliability, particularly in high-stakes or resource-constrained settings, such as medical diagnostics (e.g., Olsson et al. [2022]). This widespread applicability has spurred considerable interest in improving the informativeness of conformal sets, with the goal of providing more precise and actionable predictions.

However, traditional conformal prediction frameworks suffer from a key limitation: the miscoverage rate is predetermined, and the resulting conformal sets are not controllable. To address this, recent research has explored methods that allow for adaptive, data-dependent miscoverage rates. Notable contributions include the work of Sarkar and Kuchibhotla [2023], which proposes constructing valid coverage guarantees simultaneously for each miscoverage level with high probability. Their approach relies on constructing confidence bands for cumulative distribution functions, which is analogous to the simultaneous inference framework of post-selection inference introduced by Berk et al. [2013]. However, one limitation of this method is that it relies on the i.i.d. assumption. Similar techniques with post-hoc miscoverage have been used in various conformal prediction setups, such as in the transductive setting where multiple test scores are considered [Gazin et al., 2024], in the multiple testing setting [Wang et al., 2024], or in risk control [Nguyen et al., 2024].

Another important related direction is the work of Cherian et al. [2024], which combines two powerful ideas: conformal prediction with conditional coverage guarantees, and level-adaptive conformal prediction. In their framework, the miscoverage level $\alpha(.)$ is treated as a learned function, optimized to satisfy some quality criterion. In contrast, our goal is not to learn $\alpha(.)$, but rather to estimate the marginal guarantees that result from applying a user-specified constraint on the size of the conformal set. The level $\alpha$ in our case is directly derived to satisfy this constraint. Additionally, we allow the level $\alpha$ to depend not only on the test features but also on the calibration set, which introduces further flexibility in adapting the coverage to the data.

We propose a more direct approach based on e-values [Vovk and Wang, 2021, Grünwald et al., 2024], a robust alternative to p-values that offers several advantages such as stronger data-dependent Type-I error guarantees, which we leverage in this work. Our method yields marginal coverage guarantees, and we provide a practical procedure to estimate the corresponding marginal miscoverage term.

# B  Further experimental details

All experiments were run on a machine with a 13th Gen Intel® Core™ i7-13700H CPU and it typically takes 0.1-1.5 hours for each trial, depending on the calibration size $n$. First, we provide additional details on the experiment conducted in Section 4, followed by an illustration of Backward Conformal Prediction applied to other experiments.

## B.1  Additional details for the experiment in Section 4

For completeness, we provide samples of histograms of $\tilde{\alpha}_j$ for $j = 1, \ldots, n$ in Figures 4 and 5, which are used to compute the aggregated estimate $\hat{\alpha}^{\mathrm{LOO}}$. These samples correspond to the histograms obtained from a single run of the experiment conducted in Section 4. We use the same setup as in that section, with a constant size constraint rule $\mathcal{T} = 1$ and a calibration size of $n = 5000$.

## B.2  Introductory setting: binary classification

Now, we highlight the importance of controlling the size of prediction sets with a binary classification example.

In binary classification, conformal prediction yields prediction sets of the form $\varnothing, \{0\}, \{1\}$, or $\{0, 1\}$. However, applying standard conformal methods with a fixed miscoverage level $\alpha$ can often result in trivial prediction sets equal to $\{0, 1\}$, which offer no information. Ideally, one would prefer prediction sets of size at most $\mathcal{T} = 1$, providing decisive and actionable predictions.

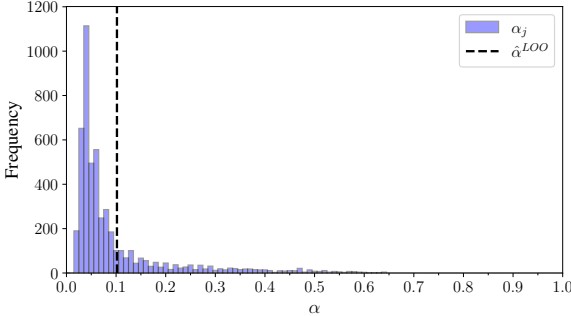

Figure 4: Sample histogram 1 of the values $\tilde{\alpha}_j$, for $j = 1, \ldots, n$, used to compute $\hat{\alpha}^{\mathrm{LOO}}$ with $\mathcal{T} = 1$ and $n = 5000$.

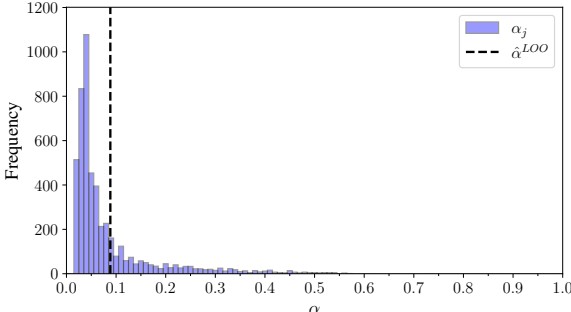

Figure 5: Sample histogram 2 of the values $\tilde{\alpha}_j$, for $j = 1, \ldots, n$, used to compute $\hat{\alpha}^{\mathrm{LOO}}$ with $\mathcal{T} = 1$ and $n = 5000$.

We perform binary classification experiments on the Breast Cancer Wisconsin (Diagnostic) dataset [Wolberg et al., 1993]. The dataset consists of 569 instances, each labeled as either benign ($y = 0$) or malignant ($y = 1$), with 30 real-valued features computed from digitized images of fine needle aspirates of breast masses.

We randomly split the dataset into 70% training and 30% testing data. We train an XGBoost classifier $f$ using cross-entropy [Chen and Guestrin, 2016]. The trained model $f$ achieves an accuracy of 97.7% on the test set.

We repeat the experiment $N = 200$ times. In each iteration, we randomly sample a calibration set $\{(X_i, Y_i)\}_{i=1}^n$, consisting of half of the original test set, drawn uniformly at random. We compute conformity scores using the cross-entropy loss, defined for binary classification as

$$S(x, y) = - \left[ y \log(f(x)) + (1 - y) \log(1 - f(x)) \right],$$

where $f(x)$ denotes the predicted probability of class 1 produced by the pre-trained model $f$.

From the remaining test points (i.e., those not included in the calibration set), we select one sample $(X_{\text{test}}, Y_{\text{test}})$ uniformly at random to serve as the test input.

When applying standard conformal prediction methods with a fixed miscoverage level, the resulting conformal sets may turn out to be trivial. For instance, over $N = 200$ runs with a fixed miscoverage level $\alpha = 0.02$, we obtain 1 empty conformal set, 166 informative sets of size 1, and 33 uninformative sets of size 2. This means that out of a total of 200 patients, 34 of them receive prediction sets that offer no insight into their diagnosis.

To address this issue, we instead use Backward Conformal Prediction with a fixed prediction set size $\mathcal{T} = 1$. Based on $X_{\text{test}}$ and the calibration set, we compute an adaptive miscoverage $\tilde{\alpha}$ following (4), and construct the corresponding conformal prediction set $\hat{C}_n^{\tilde{\alpha}}(X_{\text{test}})$ of size at most $\mathcal{T} = 1$.

The adaptive miscoverage $\tilde{\alpha}$ as described in Equation (4) is computed using binary search with tolerance 0.005. For each run, we record the corresponding value of $\tilde{\alpha}$ and the resulting prediction set $\hat{C}_n^{\tilde{\alpha}}(X_{\text{test}})$. In Figure 6, we plot the histogram of the $N$ values of $1 - \tilde{\alpha}$ obtained across runs

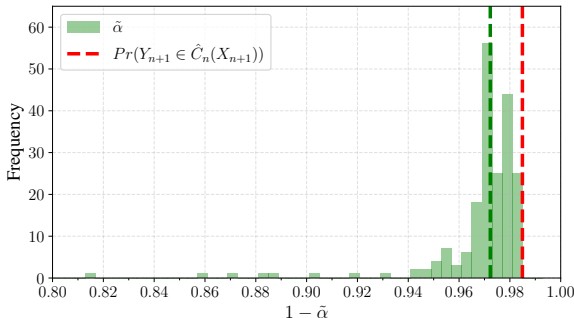

Figure 6: Plot of the adaptive miscoverage levels $\tilde{\alpha}$ and the empirical coverage rate $\mathbb{P}(Y_{\text{test}} \in \hat{C}_n^{\tilde{\alpha}}(X_{\text{test}}))$ on the binary classification task, using Backward Conformal Prediction with fixed set size $\mathcal{T} = 1$.

in green, with the mean $1 - \mathbb{E}[\tilde{\alpha}]$ shown as a green dashed line. We also report the empirical coverage probability $\mathbb{P}(Y_{\text{test}} \in \hat{C}_n^{\tilde{\alpha}}(X_{\text{test}}))$ as a red dashed line. We observe that the coverage probability $\mathbb{P}(Y_{\text{test}} \in \hat{C}_n^{\tilde{\alpha}}(X_{\text{test}}))$ consistently exceeds the marginal coverage $1 - \mathbb{E}[\tilde{\alpha}]$, providing additional empirical support for the coverage guarantee (2).

Note that even when using standard conformal prediction methods with a fixed miscoverage level set to $\alpha = \mathbb{E}[\tilde{\alpha}]$, we still obtain 196 conformal sets of size 1 and 4 sets of size 2. As a result, some patients receive prediction sets that provide no informative guidance. This simple binary classification example highlights the need for adaptive control of prediction set sizes, which can be addressed using Backward Conformal Prediction.

### B.2.1 Adaptive size constraint rule

We also illustrate Backward Conformal Prediction with a size constraint rule that adapts to the data. Intuitively, features associated with high label variability, as estimated on the calibration set, correspond to greater uncertainty and should be assigned larger prediction sets. We define the size constraint rule $\mathcal{T}$ in a data-dependent manner, adapting the desired prediction set size to the local uncertainty around each test point.

We begin by extracting features from a pretrained ResNet-18 network [He et al., 2016], and project them onto a two-dimensional space using principal component analysis (PCA); for a comprehensive treatment of PCA, see Jolliffe [2002]. For each $X_i$ in the calibration set, we identify its $k$ nearest neighbors in the PCA space, denoted $\mathcal{N}_k(X_i)$, and collect the corresponding labels $\{Y_j : j \in \mathcal{N}_k(X_i)\}$.

We then compute the empirical label distribution in the neighborhood of $X_i$,

$$\hat{p}_{X_i}(c) = \frac{1}{k} \sum_{j \in \mathcal{N}_k(X_i)} \mathbb{1}\{Y_j = c\}, \quad \text{for } c \in \{1, \dots, |\mathcal{Y}|\},$$

and define the local label entropy as

$$H(X_i) = -\sum_{c=1}^{|\mathcal{Y}|} \hat{p}_{X_i}(c) \log_2 \hat{p}_{X_i}(c).$$

This yields a scalar uncertainty score for each calibration point. We compute $H(X_{\text{test}})$ similarly for any test point $X_{\text{test}}$, by considering its $k$ nearest neighbors among the calibration features.

To map the local entropy to a discrete prediction set size, we define a binning function $b : \mathbb{R} \to \{\mathcal{T}_{\min}, \dots, \mathcal{T}_{\max}\}$, where $\mathcal{T}_{\min}$ and $\mathcal{T}_{\max}$ denote the minimum and maximum allowable sizes for the prediction sets.

Given the entropy value $H(X_{\text{test}})$ around a test point $X_{\text{test}}$, the size constraint rule is defined as:

$$\mathcal{T}(\{(X_i, Y_i)\}_{i=1}^n, X_{\text{test}}) = b(H(X_{\text{test}})). \tag{6}$$

To construct $b$, we partition the range of entropy values observed on the calibration set into $L = \mathcal{T}_{\max} - \mathcal{T}_{\min} + 1$ bins. These bins are defined by thresholds:

$$\text{bins} = \left\{ \min_{i=1,\dots,n} H(X_i) + \left( \max_{i=1,\dots,n} H(X_i) - \min_{i=1,\dots,n} H(X_i) \right) \cdot \left( \frac{\ell - 1}{L - 1} \right)^{1/2} \right\}_{\ell=1}^{L},$$

where the exponent $1/2$ is arbitrary and skews the binning towards low entropy regions.

The binning function $b$ then maps an entropy value $h$ to a size $\mathcal{T}_{\min} + \ell$, where $\ell$ is the index of the bin containing $h$. That is,

$$b(h) = \mathcal{T}_{\min} + \sum_{\ell=1}^{L-1} \mathbb{1}\{h > \text{bins}_\ell\}.$$

In summary, $\mathcal{T}(\{(X_i, Y_i)\}_{i=1}^n, X_{\text{test}})$ is larger when the local neighborhood of $X_{\text{test}}$ exhibits high label variability, leading to broader prediction sets in more ambiguous regions of the input space.

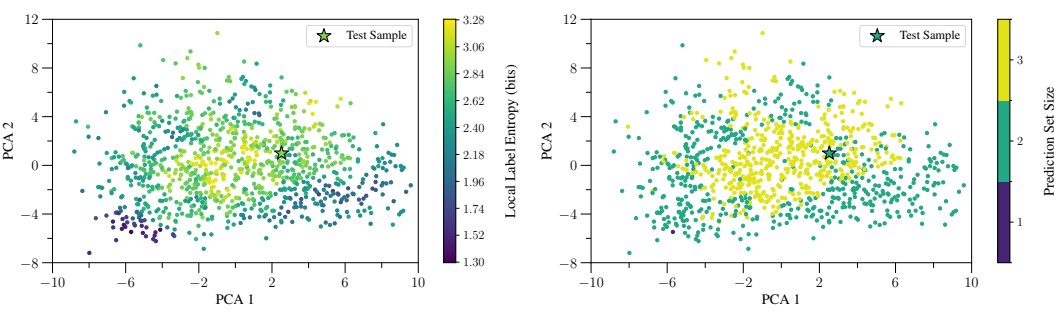

Local entropy of test point: 2.648.  Prediction set size for the test sample: 2.

Figure 7: Illustration of the size constraint rule on the calibration and test samples. **Left:** local label entropy in PCA space. **Right:** corresponding values of the size constraint rule $\mathcal{T}$ defined in Equation (6) applied to both calibration and test points.

We repeat the same experiment as in the previous subsection with constant $\mathcal{T}$, except this time we use the adaptive, data-dependent rule defined above with $\mathcal{T}_{\min} = 1$, $\mathcal{T}_{\max} = 3$, and $k = 20$. An illustration of the adaptive size constraint rule $\mathcal{T}$ defined in Equation (6) with $n = 1000$ is provided in Figure 7. The corresponding coverages are presented in Figure 8.

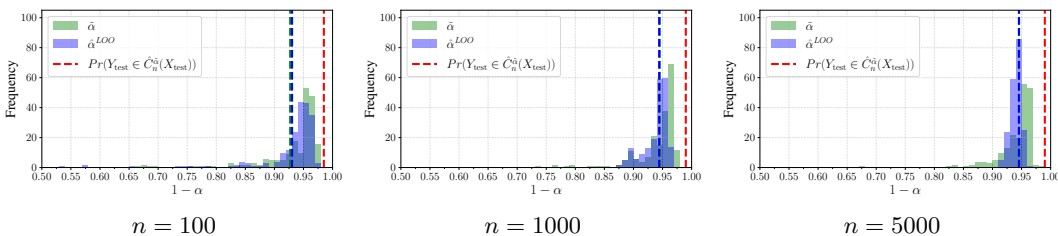

$n = 100$ $\qquad\qquad\qquad n = 1000 \qquad\qquad\qquad n = 5000$

Figure 8: Results for the adaptive $\mathcal{T}$ defined in Equation (6).

We observe that the coverage probability exceeds the empirical coverage, supporting the validity of (2). Additionally, as in the constant case, the estimator $\hat{\alpha}^{\text{LOO}}$ increasingly approximates the marginal miscoverage $\mathbb{E}[\tilde{\alpha}]$ as $n$ grows, in line with Theorem 3.5. These observations suggest that Backward Conformal Prediction provides a principled approach to uncertainty quantification in machine learning yielding controlled-size prediction sets, even when the set size is dynamically adapted based on the data.

# C Proofs for Section 3

## C.1 Constant size constraint rule

Properties (P1) and (P2) allow us to restrict the analysis to the open subset of $\mathbb{R}_+^{|\mathcal{Y}|}$ defined by

$$\mathbb{R}_+^{|\mathcal{Y}|,[1,\mathcal{T}]} := \left\{ \mathbf{E} = (\mathbf{E}_y)_{y\in\mathcal{Y}} \in \mathbb{R}_+^{|\mathcal{Y}|} \mid 1 \le \# \{y : \mathbf{E}_y < 1\} \le \mathcal{T}, \ \mathbf{E}_y \ne 1 \ \forall y \in \mathcal{Y} \right\}$$

$$= \bigcup_{\substack{I \subseteq \{1,\ldots,|\mathcal{Y}|\} \\ 1 \le |I| \le \mathcal{T}}} \left\{ \mathbf{E} = (\mathbf{E}_y)_{y\in\mathcal{Y}} \in \mathbb{R}_+^{|\mathcal{Y}|} \mid \mathbf{E}_i < 1 \ \forall i \in I, \ \mathbf{E}_j > 1 \ \forall j \notin I \right\},$$

which is an open set as an union of finite intersections of open subsets of $\mathbb{R}_+^{|\mathcal{Y}|}$. In the following, we denote by $\|.\|$ the infinity norm on $\mathbb{R}_+^{|\mathcal{Y}|,[1,\mathcal{T}]}$.

We introduce:

$$F : \begin{array}{rcl} \mathbb{R}_+^{|\mathcal{Y}|,[1,\mathcal{T}]} & \longrightarrow & [0,1] \\ \mathbf{E} = (\mathbf{E}_y)_{y\in\mathcal{Y}} & \longmapsto & \inf \{\alpha \in (0,1) : \# \{y : \mathbf{E}_y < 1/\alpha\} \le \mathcal{T}\} \end{array} \tag{7}$$

as a tool to study the behavior of $\tilde{\alpha}$ defined in (4).

### C.1.1 Stability of $F$

In the definition of $F$, the cardinality $\# \{y : \mathbf{E}_y < 1/\alpha\}$ is a sum of indicator functions: $\sum_y \mathbb{1}_{\{\mathbf{E}_y < 1/\alpha\}}$ which is non-differentiable. For $\lambda > 0$, the bump: $\sigma(\lambda(1/\alpha - \mathbf{E}_y))$, where:

$$\sigma(x) = \begin{cases} \exp\left(-\frac{1}{x}\right) & \text{if } x > 0, \\ 0 & \text{if } x \le 0, \end{cases} \tag{8}$$

acts as a soft indicator: it is close to 1 when the inequality $\mathbf{E}_y < 1/\alpha$ is satisfied and close to 0 otherwise. Note that

$$\sigma'(x) = \begin{cases} \frac{1}{x^2} \exp\left(-\frac{1}{x}\right) & \text{if } x > 0, \\ 0 & \text{if } x \le 0. \end{cases}$$

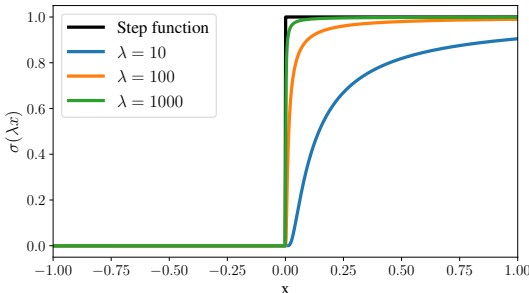

Figure 9: Visualization of the smooth approximation $\sigma(\lambda x)$ to the step function for various values of $\lambda$. As $\lambda$ increases, the transition from 0 to 1 becomes sharper, making $\sigma(\lambda x)$ a closer approximation to the discontinuous step function.

In our analysis, we approximate the non-differentiable cardinality condition using a smooth surrogate by replacing the indicator function with this bump approximation. Specifically, we consider:

$$F_\lambda : \begin{array}{rcl} \mathbb{R}_+^{|\mathcal{Y}|,[1,\mathcal{T}]} & \longrightarrow & [0,1] \\ \mathbf{E} = (\mathbf{E}_y)_{y\in\mathcal{Y}} & \longmapsto & \inf \left\{\alpha \in (0,1) : \sum_{y\in\mathcal{Y}} \sigma(\lambda(1/\alpha - \mathbf{E}_y)) \le \mathcal{T}\right\} \end{array} \tag{9}$$

for $\lambda > 0$. This replacement ensures differentiability with respect to $\alpha$, and recovers the original cardinality condition in the limit $\lambda \to \infty$.

We now show that this smooth surrogate is meaningful by proving that the associated infimum $F_\lambda$ converges pointwise to $F$ as $\lambda \to \infty$.

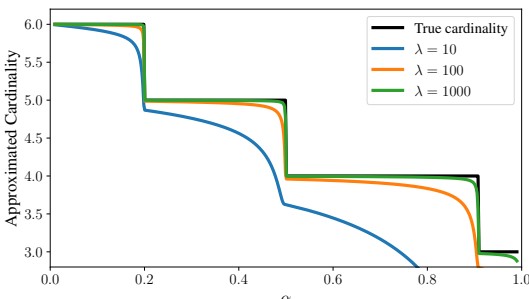

Figure 10: Illustration of the difference between $\#\{y : \mathbf{E}_y < 1/\alpha\}$ (in black) and its smooth approximation $\sigma(\lambda(1/\alpha - \mathbf{E}_y))$. As $\lambda$ increases, $\sigma(\lambda(1/\alpha - \mathbf{E}_y))$ recovers the original cardinality.

**Lemma C.1.** *Let* $\mathbf{E} \in \mathbb{R}_+^{|\mathcal{Y}|,[1,\mathcal{T}]}$. *Then,*

$$\lim_{\lambda \to \infty} F_\lambda(\mathbf{E}) = F(\mathbf{E}).$$

*Proof.* Define the following two functions for any $\alpha \in (0,1)$:

$$f(\mathbf{E}, \alpha) := \#\{y : \mathbf{E}_y < 1/\alpha\} = \sum_{y \in \mathcal{Y}} \mathbb{1}_{\{\mathbf{E}_y < 1/\alpha\}},$$

$$f_\lambda(\mathbf{E}, \alpha) := \sum_{y \in \mathcal{Y}} \sigma(\lambda(1/\alpha - \mathbf{E}_y)),$$

where $\sigma$ is the bump function defined in (8).

By construction, we have for all $\lambda > 0$ and $\alpha \in (0,1)$:

$$f_\lambda(\mathbf{E}, \alpha) \leq f(\mathbf{E}, \alpha),$$

since $\sigma(\lambda(1/\alpha - \mathbf{E}_y)) \leq 1$ and vanishes when $\mathbf{E}_y \geq 1/\alpha$. It follows that:

$$F_\lambda(\mathbf{E}) \leq F(\mathbf{E}).$$

Now, let $\varepsilon > 0$. We will show that $F_\lambda(\mathbf{E}) \geq F(\mathbf{E}) - \varepsilon$ when $\lambda$ is sufficiently large. Assume $F(\mathbf{E}) > 0$ (otherwise, the result is trivial). Define:

$$\alpha_\varepsilon := F(\mathbf{E}) - \varepsilon,$$

and assume, without loss of generality, that $\varepsilon$ is small enough so that $\alpha_\varepsilon \in (0,1)$. By definition of $F(\mathbf{E})$, we have:

$$f(\mathbf{E}, \alpha_\varepsilon) > \mathcal{T},$$

because $\alpha_\varepsilon < F(\mathbf{E})$. Moreover, since $f_\lambda$ converges pointwise to $f$ as $\lambda \to \infty$ by definition of the bump function $\sigma$, there exists $\lambda_0 > 0$ such that for all $\lambda \geq \lambda_0$,

$$f_\lambda(\mathbf{E}, \alpha_\varepsilon^-) > \mathcal{T}.$$

This means that $\alpha_\varepsilon$ is not feasible in the infimum defining $F_\lambda(\mathbf{E})$, and since $f_\lambda$ is non-increasing in its second argument:

$$F_\lambda(\mathbf{E}) \geq \alpha_\varepsilon = F(\mathbf{E}) - \varepsilon \quad \text{for all } \lambda \geq \lambda_0.$$

Combining both bounds, for all $\lambda \geq \lambda_0$,

$$F(\mathbf{E}) - \varepsilon \leq F_\lambda(\mathbf{E}) \leq F(\mathbf{E}).$$

Since this holds for any $\varepsilon > 0$, we conclude:

$$\lim_{\lambda \to \infty} F_\lambda(\mathbf{E}) = F(\mathbf{E}).$$

$\square$

Now, we bound the differences between the values of $F_\lambda$ as the inputs vary, and we will prove that $F_\lambda$ is 1-Lipschitz on any compact set $\mathcal{K}$ of $\mathbb{R}_+^{|\mathcal{Y}|,[1,\mathcal{T}]}$. This result naturally follows from the Implicit Function Theorem, a standard result in analysis (see, for instance, Rudin [1953]).

**Lemma C.2.** *Let $\mathcal{K}$ be a compact subset of $\mathbb{R}_+^{|\mathcal{Y}|,[1,C]}$. Assume that $\mathcal{T} < |\mathcal{Y}|$. Then there exists $\bar{\lambda}$ such that for all $\lambda \geq \bar{\lambda}$, there exists a constant $k_\lambda$ such that:*

$$|F_\lambda(\mathbf{E}^1) - F_\lambda(\mathbf{E}^2)| \leq k_\lambda \|\mathbf{E}^1 - \mathbf{E}^2\|,$$

*for all $\mathbf{E}^1, \mathbf{E}^2 \in \mathcal{K}$. In other words, $F_\lambda$ is Lipschitz continuous on $\mathcal{K}$. Moreover, $k_\lambda \leq 1$.*

*Proof.* Let $\lambda > 0$. To show the existence of $k_\lambda$, it suffices to prove that $F_\lambda$ is continuously differentiable on an open set $U$ of $\mathbb{R}_+^{|\mathcal{Y}|,[1,\mathcal{T}]}$ containing $\mathcal{K}$, since by the Mean Value Theorem, a continuously differentiable function on a compact set is Lipschitz continuous. We fix such an open set $U$ in the following.

To apply the Implicit Function Theorem, we may first handle the singularity in $\alpha = 0$. We assume without loss of generality that:

$$U = \left\{ \mathbf{E} = (\mathbf{E}_y)_{y \in \mathcal{Y}} \in \mathbb{R}_+^{|\mathcal{Y}|,[1,\mathcal{T}]} \mid \|\mathbf{E}\| < M \right\}$$

for some constant $M > 0$, which is possible since $\mathcal{K}$ is bounded. Let $\alpha_M := \frac{1}{M+1} > 0$, and let $\bar{\lambda} := \frac{1}{\log(|\mathcal{Y}|/\mathcal{T})}$ which is $> 0$ since $\mathcal{T} < |\mathcal{Y}|$. For all $\mathbf{E} \in U$ and $\lambda \geq \bar{\lambda}$, we have:

$$\sum_{y \in \mathcal{Y}} \sigma(\lambda(1/\alpha_M - \mathbf{E}_y)) = \sum_{y \in \mathcal{Y}} \sigma(\lambda(M + 1 - \mathbf{E}_y))$$

$$> \sum_{y \in \mathcal{Y}} \sigma(\lambda)$$

$$= |\mathcal{Y}| \, e^{-\frac{1}{\lambda}}$$

$$\geq \mathcal{T},$$

where the first inequality comes from the definition of $U$ and the second from the definition of $\bar{\lambda}$. Thus, for all $\mathbf{E} \in U$ and $\lambda \geq \bar{\lambda}$, we have $F_\lambda(\mathbf{E}) \geq \alpha_M > 0$, allowing us to restrict our analysis to $(0, 1]$ instead of $[0, 1]$.

In the following, we fix $\lambda \geq \bar{\lambda}$.

Define:

$$\Phi_\lambda(\mathbf{E}, \alpha) := \sum_{y \in \mathcal{Y}} \sigma(\lambda(1/\alpha - \mathbf{E}_y)) - \mathcal{T},$$

for $\mathbf{E} = (\mathbf{E}_y)_{y \in \mathcal{Y}} \in U$ and $\alpha \in (0, 1]$. The function $\Phi_\lambda$ is continuously differentiable. Fix $\mathbf{E}^0 \in U$. For any $\alpha \in (0, 1]$, we have:

$$\frac{\partial \Phi_\lambda}{\partial \alpha}(\mathbf{E}^0, \alpha) = \frac{-\lambda}{\alpha^2} \sum_{y \in \mathcal{Y}} \sigma'(\lambda(1/\alpha - \mathbf{E}_y^0)) < 0. \tag{10}$$

The inequality holds because $\sigma' \geq 0$, and since $\mathbf{E}^0 \in \mathbb{R}_+^{|\mathcal{Y}|,[1,\mathcal{T}]}$, there exists at least one $y \in \mathcal{Y}$ such that $\mathbf{E}_y^0 < 1$. For such a $y$, we have $1/\alpha > 1 > \mathbf{E}_y^0$, hence $\sigma'(\lambda(1/\alpha - \mathbf{E}_y^0)) > 0$, which makes the sum strictly positive.

Moreover, for $\mathbf{E} \in U$, $F_\lambda(\mathbf{E})$ is the unique solution to the equation $\Phi_\lambda(\mathbf{E}, F_\lambda(\mathbf{E})) = 0$ by the Intermediate Value Theorem. Indeed, $\Phi_\lambda(\mathbf{E}, .)$ is continuous on $(0, 1]$ and strictly decreasing with $\Phi_\lambda(\mathbf{E}, \alpha_M) > 0$ and $\Phi_\lambda(\mathbf{E}, 1) = \sum_{y \in \mathcal{Y}} \sigma(\lambda(1 - \mathbf{E}_y)) - \mathcal{T} < \#\{y : \mathbf{E}_y < 1\} - \mathcal{T} \leq 0$, where the strict inequality holds because there exists at least one $y \in \mathcal{Y}$ such that $\mathbf{E}_y < 1$.

By the Implicit Function Theorem, there exists a neighborhood $U_0$ of $\mathbf{E}^0$ and a unique continuously differentiable function $\phi_\lambda : U_0 \to (0, 1]$ such that $\phi_\lambda(\mathbf{E}^0) = F_\lambda(\mathbf{E}^0)$ and $\Phi_\lambda(\mathbf{E}, \phi_\lambda(\mathbf{E})) = 0$ for all $\mathbf{E} \in U_0$. By uniqueness, we have $F_\lambda = \phi_\lambda$ on $U_0$, so $F_\lambda$ is continuously differentiable on $U_0$. As this holds for any $\mathbf{E}^0 \in U$, we conclude that $F_\lambda$ is continuously differentiable on $U$.

The Mean Value Theorem guarantees that $F_\lambda$ is Lipschitz continuous on $\mathcal{K}$, with the Lipschitz constant given by $k_\lambda = \sup_{\mathbf{E} \in \mathcal{K}} \|\nabla F_\lambda(\mathbf{E})\|$. It remains to show that $k_\lambda \leq 1$.

By the Implicit Function Theorem, we also know that:

$$\nabla F_\lambda(\mathbf{E}) = -\frac{1}{\frac{\partial \Phi_\lambda}{\partial \alpha}} \nabla_\mathbf{E} \Phi_\lambda(\mathbf{E}, F_\lambda(\mathbf{E})),$$

where $\frac{\partial \Phi_\lambda}{\partial \alpha}$ is given in (10), and

$$\nabla_{\mathbf{E}} \Phi_\lambda(\mathbf{E}, F_\lambda(\mathbf{E})) = \left( -\lambda \sigma' \left( \lambda \left( \frac{1}{F_\lambda(\mathbf{E})} - \mathbf{E}_y \right) \right) \right)_{y \in \mathcal{Y}}.$$

Thus, we can express the norm of the gradient as:

$$\|\nabla F_\lambda(\mathbf{E})\| = \max_{y \in \mathcal{Y}} \left| \frac{-\lambda \sigma' \left( \lambda \left( \frac{1}{F_\lambda(\mathbf{E})} - \mathbf{E}_y \right) \right)}{\frac{-\lambda}{F_\lambda(\mathbf{E})^2} \sum_{y \in \mathcal{Y}} \sigma' \left( \lambda \left( \frac{1}{F_\lambda(\mathbf{E})} - \mathbf{E}_y \right) \right)} \right|$$

$$= F_\lambda(\mathbf{E})^2 \frac{\max_{y \in \mathcal{Y}} \sigma' \left( \lambda \left( \frac{1}{F_\lambda(\mathbf{E})} - \mathbf{E}_y \right) \right)}{\sum_{y \in \mathcal{Y}} \sigma' \left( \lambda \left( \frac{1}{F_\lambda(\mathbf{E})} - \mathbf{E}_y \right) \right)}$$

$$\leq F_\lambda(\mathbf{E})^2$$

$$\leq 1,$$

where the first inquality comes from the fact that $\sigma' \geq 0$, and the second inequality follows from the fact that $F_\lambda(\mathbf{E}) \in (0, 1)$. Therefore, $\|\nabla F_\lambda(\mathbf{E})\| \leq 1$. Finally, we conclude that $k_\lambda \leq 1$. □

As a direct consequence of Lemmas C.1 and C.2, we state the following stability result.

**Theorem C.3.** *Let $\mathcal{K}$ be a compact subset of $\mathbb{R}_+^{|\mathcal{Y}|, [1, \mathcal{T}]}$. Assume that $\mathcal{T} < |\mathcal{Y}|$. Then the function $F$ is 1-Lipschitz on $\mathcal{K}$, that is,*

$$|F(\mathbf{E}^1) - F(\mathbf{E}^2)| \leq \|\mathbf{E}^1 - \mathbf{E}^2\|,$$

*for all $\mathbf{E}^1, \mathbf{E}^2 \in \mathcal{K}$.*

### C.1.2 Properties of $\hat{\alpha}^{\mathrm{LOO}}$

We now show how the stability property of $F$ can be leveraged to derive key properties of the estimator $\hat{\alpha}^{\mathrm{LOO}}$, which approximates the miscoverage term $\mathbb{E}[\tilde{\alpha}]$, where $\tilde{\alpha}$ was defined in (4).

**Theorem C.4** (Theorem 3.1). *Assume that the score function $S$ is bounded and takes values in the interval $[S_{\min}, S_{\max}]$, with $0 < S_{\min} \leq S_{\max}$. Suppose that the number of calibration samples satisfies $n > S_{\max}/S_{\min}$. In addition, assume that the vectors $\mathbf{E}^{\mathrm{test}}$, $\mathbf{E}^j$, $\tilde{\mathbf{E}}^{\mathrm{test}}$, and $\tilde{\mathbf{E}}^j$ satisfy the following properties almost surely:*

*(P1)* $1 \leq \# \{y \in \mathcal{Y} \mid \mathbf{E}_y < 1\} \leq \mathcal{T} < |\mathcal{Y}|$;

*(P2) For all $y \in \mathcal{Y}$, $\mathbf{E}_y \neq 1$.*

*Then, if the samples $(X_i, Y_i)$ are i.i.d., the leave-one-out estimator satisfies:*

$$\left| \hat{\alpha}^{\mathrm{LOO}} - \mathbb{E}[\tilde{\alpha}] \right| = O_P \left( \frac{1}{\sqrt{n}} \right).$$

*Proof.* First, since the vectors $\mathbf{E}^{\mathrm{test}}$, $\mathbf{E}^j$, $\tilde{\mathbf{E}}^{\mathrm{test}}$, and $\tilde{\mathbf{E}}^j$ satisfy Properties (P1) and (P2), they lie in $\mathbb{R}_+^{|\mathcal{Y}|, [1, \mathcal{T}]}$ almost surely. Moreover, since $S$ is bounded, these vectors belong to a compact subset of $\mathbb{R}_+^{|\mathcal{Y}|, [1, \mathcal{T}]}$, which allows us to apply Theorem C.3.

By the triangular inequality, we have:

$$\left| \hat{\alpha}^{\mathrm{LOO}} - \mathbb{E}[\tilde{\alpha}] \right| = \left| \frac{1}{n} \sum_{j=1}^n F(\mathbf{E}^j) - \mathbb{E}[F(\mathbf{E}^{\mathrm{test}})] \right|$$

$$= \left| \frac{1}{n} \sum_{j=1}^n \left( F(\mathbf{E}^j) - F\left(\tilde{\mathbf{E}}^j\right) \right) + \left( F\left(\tilde{\mathbf{E}}^j\right) - \mathbb{E}\left[F\left(\tilde{\mathbf{E}}^{\mathrm{test}}\right)\right] \right) + \left( \mathbb{E}\left[F\left(\tilde{\mathbf{E}}^{\mathrm{test}}\right)\right] - \mathbb{E}[F(\mathbf{E}^{\mathrm{test}})] \right) \right|$$

$$\leq \underbrace{\left| \frac{1}{n} \sum_{j=1}^n \left( F(\mathbf{E}^j) - F\left(\tilde{\mathbf{E}}^j\right) \right) \right|}_{=: T_1} + \underbrace{\left| \frac{1}{n} \sum_{j=1}^n \left( F\left(\tilde{\mathbf{E}}^j\right) - \mathbb{E}\left[F\left(\tilde{\mathbf{E}}^{\mathrm{test}}\right)\right] \right) \right|}_{=: T_2} + \underbrace{\left| \mathbb{E}\left[F\left(\tilde{\mathbf{E}}^{\mathrm{test}}\right)\right] - \mathbb{E}[F(\mathbf{E}^{\mathrm{test}})] \right|}_{=: T_3}.$$

We will now proceed to bound each of these terms individually, establishing high probability bounds for $T_1$ and $T_2$. Let $\delta > 0$.

We start with the expression for $T_1$:

$$T_1 = \left| \frac{1}{n} \sum_{j=1}^{n} \left( F(\mathbf{E}^j) - F\left( \tilde{\mathbf{E}}^j \right) \right) \right|.$$

By applying the triangle inequality, we have:

$$T_1 \leq \frac{1}{n} \sum_{j=1}^{n} \left| F(\mathbf{E}^j) - F\left( \tilde{\mathbf{E}}^j \right) \right|.$$

Using the Lipschitz continuity of $F$ (Theorem C.3), we can bound each term as follows:

$$T_1 \leq \frac{1}{n} \sum_{j=1}^{n} \left\| \mathbf{E}^j - \tilde{\mathbf{E}}^j \right\|.$$

Now, for each $j \in \{1, \cdots, n\}$ and $y \in \mathcal{Y}$, we examine the difference between the components of $\mathbf{E}^j$ and $\tilde{\mathbf{E}}^j$:

$$\left| \mathbf{E}_y^j - \tilde{\mathbf{E}}_y^j \right| = \left| \frac{S(X_j, y)}{\frac{1}{n} \left( \sum_{i=1, i \neq j}^{n} S(X_i, Y_i) + S(X_j, y) \right)} - \frac{S(X_j, y)}{\mu} \right|.$$

This can be bounded as:

$$\leq S_{\max} \left| \frac{1}{\frac{1}{n} \left( \sum_{i=1, i \neq j}^{n} S(X_i, Y_i) + S(X_j, y) \right)} - \frac{1}{\mu} \right|.$$

Simplifying the expression inside the absolute value:

$$= S_{\max} \left| \frac{\mu - \frac{1}{n} \sum_{i=1, i \neq j}^{n} S(X_i, Y_i) - \frac{S(X_j, y)}{n}}{\mu \left( \frac{1}{n} \sum_{i=1, i \neq j}^{n} S(X_i, Y_i) + \frac{S(X_j, y)}{n} \right)} \right|.$$

We continue simplifying:

$$= S_{\max} \frac{\left| \mu - \frac{1}{n} \sum_{i=1}^{n} S(X_i, Y_i) + \frac{S(X_j, Y_j)}{n} - \frac{S(X_j, y)}{n} \right|}{\mu \left( \frac{1}{n} \sum_{i=1}^{n} S(X_i, Y_i) - \frac{S(X_j, Y_j)}{n} + \frac{S(X_j, y)}{n} \right)}.$$

We now apply Hoeffding's inequality to the numerator and use the bounds on $S$:

$$\leq S_{\max} \frac{\sqrt{\frac{S_{\max}^2 \log(2/\delta)}{2n}} + \frac{2 S_{\max}}{n}}{\mu \left( S_{\min} - \frac{S_{\max}}{n} \right)}.$$

Finally, simplifying further:

$$= S_{\max} \frac{\sqrt{\frac{\log(2/\delta)}{2n}} + \frac{2}{n}}{\mu \left( S_{\min}/S_{\max} - \frac{1}{n} \right)}.$$

Thus, we obtain the following bound for $T_1$:

$$T_1 \leq S_{\max} \frac{\sqrt{\frac{\log(2/\delta)}{2n}} + \frac{2}{n}}{\mu \left( S_{\min}/S_{\max} - \frac{1}{n} \right)} \quad \text{with probability} \geq 1 - \delta.$$

Next, we bound $T_2$. Since the $\tilde{\mathbf{E}}^j$ are i.i.d. random variables, we can apply Hoeffding's inequality. Specifically, we have:

$$T_2 \leq \sqrt{\frac{\log(2/\delta)}{2n}} \quad \text{with probability} \geq 1 - \delta,$$

because $F$ takes values in the interval $[0, 1]$.

Finally, we bound $T_3$. We begin by rewriting $T_3$ as follows:

$$T_3 = \left| \mathbb{E}[F(\tilde{\mathbf{E}}^{\text{test}})] - \mathbb{E}[F(\mathbf{E}^{\text{test}})] \right| = \left| \mathbb{E}\left[ F(\tilde{\mathbf{E}}^{\text{test}}) - F(\mathbf{E}^{\text{test}}) \right] \right|.$$

By the triangle inequality, we can express this as:

$$T_3 \leq \mathbb{E}\left[ \left| F(\tilde{\mathbf{E}}^{\text{test}}) - F(\mathbf{E}^{\text{test}}) \right| \right].$$

Now, applying the Lipschitz condition on $F$ (from Theorem C.3), we obtain:

$$T_3 \leq \mathbb{E}\left[ \left\| \tilde{\mathbf{E}}^{\text{test}} - \mathbf{E}^{\text{test}} \right\| \right].$$

Next, we examine the difference between the components of $\tilde{\mathbf{E}}^{\text{test}}$ and $\mathbf{E}^{\text{test}}$ for each $y \in \mathcal{Y}$:

$$\left| \tilde{\mathbf{E}}_y^{\text{test}} - \mathbf{E}_y^{\text{test}} \right| = \left| \frac{S(X_{\text{test}}, y)}{\mu} - \frac{S(X_{\text{test}}, y)}{\frac{1}{n+1}\left( \sum_{i=1}^{n} S(X_i, Y_i) + S(X_{\text{test}}, y) \right)} \right|.$$

Simplifying this, we get:

$$\leq S_{\max} \left| \frac{1}{\mu} - \frac{1}{\frac{1}{n+1}\left( \sum_{i=1}^{n} S(X_i, Y_i) + S(X_{\text{test}}, y) \right)} \right|.$$

This expression can be further rewritten as:

$$= S_{\max} \left| \frac{\mu - \frac{1}{n+1}\sum_{i=1}^{n} S(X_i, Y_i) - \frac{S(X_{\text{test}}, y)}{n+1}}{\mu \left( \frac{1}{n+1}\sum_{i=1}^{n} S(X_i, Y_i) + \frac{S(X_{\text{test}}, y)}{n+1} \right)} \right|.$$

We bound the numerator and denominator separately:

$$\leq S_{\max} \frac{\left| \mu - \frac{1}{n+1}\sum_{i=1}^{n} S(X_i, Y_i) \right| + \frac{S_{\max}}{n+1}}{\mu \frac{n}{n+1} S_{\min}}.$$

We apply Hoeffding's inequality to the sum $\frac{1}{n+1}\sum_{i=1}^{n} S(X_i, Y_i)$ and obtain:

$$\leq S_{\max} \frac{\sqrt{\frac{S_{\max}^2 \log(2/\delta)}{2(n+1)}} + \frac{S_{\max}}{n+1}}{\mu \frac{n}{n+1} S_{\min}}.$$

Thus, we have:

$$\leq \frac{S_{\max}^2}{S_{\min}} \frac{\sqrt{\frac{\log(2/\delta)}{2(n+1)}} + \frac{1}{n+1}}{\mu \frac{n}{n+1}}.$$

Therefore, we conclude that:

$$\left\| \tilde{\mathbf{E}}^{\text{test}} - \mathbf{E}^{\text{test}} \right\| \leq \frac{S_{\max}^2}{S_{\min}} \frac{\sqrt{\frac{\log(2/\delta)}{2(n+1)}} + \frac{1}{n+1}}{\mu \frac{n}{n+1}} \quad \text{with probability} \geq 1 - \delta.$$

Solving for $\delta$, this means that for all $t \geq 0$:

$$\mathbb{P}\left( \left\| \tilde{\mathbf{E}}^{\text{test}} - \mathbf{E}^{\text{test}} \right\| \geq t \right) \leq 2 \exp\left( -2(n+1)\left( t\mu \frac{n}{n+1} \frac{S_{\min}}{S_{\max}^2} - \frac{1}{n+1} \right)^2 \right),$$

and hence:

$$T_3 \leq \mathbb{E}\left[ \left\| \tilde{\mathbf{E}}^{\text{test}} - \mathbf{E}^{\text{test}} \right\| \right]$$

$$= \int_0^\infty \mathbb{P}\left( \left\| \tilde{\mathbf{E}}^{\text{test}} - \mathbf{E}^{\text{test}} \right\| \geq t \right) dt$$

$$\leq \int_0^\infty 2 \exp\left( -2(n+1)\left( t\mu \frac{n}{n+1} \frac{S_{\min}}{S_{\max}^2} - \frac{1}{n+1} \right)^2 \right) dt.$$

Substituting:

$$u = t\mu \frac{n}{n+1}\frac{S_{\min}}{S_{\max}^2} - \frac{1}{n+1}, \ du = \mu\frac{n}{n+1}\frac{S_{\min}}{S_{\max}^2}dt,$$

we obtain:

$$T_3 \leq \frac{2S_{\max}^2}{\mu S_{\min}}\frac{n+1}{n}\int_{-\frac{1}{n+1}}^{\infty}\exp\left(-2(n+1)u^2\right)\,du$$

$$\leq \frac{2S_{\max}^2}{\mu S_{\min}}\frac{n+1}{n}\int_{-\infty}^{\infty}\exp\left(-2(n+1)u^2\right)\,du$$

$$= \frac{2S_{\max}^2}{\mu S_{\min}}\frac{n+1}{n}\sqrt{\frac{\pi}{2(n+1)}}.$$

Finally, by applying union bound, we obtain:

$$|\hat{\alpha}^{\mathrm{LOO}} - \mathbb{E}[\tilde{\alpha}]| \leq S_{\max}\frac{\sqrt{\frac{\log(4/\delta)}{2n}} + \frac{2}{n}}{\mu\left(S_{\min}/S_{\max} - \frac{1}{n}\right)} + \sqrt{\frac{\log(4/\delta)}{2n}} + \frac{2S_{\max}^2}{\mu S_{\min}}\frac{n+1}{n}\sqrt{\frac{\pi}{2(n+1)}},$$

that holds with probability $\geq 1 - \delta$. This shows that:

$$\left|\hat{\alpha}^{\mathrm{LOO}} - \mathbb{E}[\tilde{\alpha}]\right| = O_P\left(\frac{1}{\sqrt{n}}\right).$$

$\square$

Note that when we applied Hoeffding's concentration inequality to the sums of $S(X_i, Y_i)$, we could have used $(S_{\max} - S_{\min})^2$ instead of $S_{\max}^2$ in the error term, given that $S$ takes values in the interval $[S_{\min}, S_{\max}]$. However, we chose to keep $S_{\max}^2$ in order to keep the error terms as simple as possible.

**Theorem C.5** (Theorem 3.3). *Under the assumptions of Theorem 3.1, we have:*

$$\mathrm{Var}\left(\hat{\alpha}^{\mathrm{LOO}}\right) = O\left(\frac{1}{n}\right).$$

*Proof.* We use the same notations as in the proof of Theorem 3.1. We begin by decomposing the leave-one-out estimator $\hat{\alpha}^{\mathrm{LOO}} = \frac{1}{n}\sum_{j=1}^{n}F(\mathbf{E}^j)$ into a sum of i.i.d. components and a deviation term:

$$\hat{\alpha}^{\mathrm{LOO}} = \underbrace{\frac{1}{n}\sum_{j=1}^{n}F\left(\tilde{\mathbf{E}}^j\right)}_{=:\ \hat{\alpha}^{\mathrm{i.i.d.}}} + \underbrace{\frac{1}{n}\sum_{j=1}^{n}\left(F(\mathbf{E}^j) - F\left(\tilde{\mathbf{E}}^j\right)\right)}_{=:\ \Delta},$$

where the $F\left(\tilde{\mathbf{E}}^j\right)$ are i.i.d. copies. Since $F$ takes values in $[0, 1]$, the variance of each i.i.d. term is bounded by $\mathrm{Var}\left(F\left(\tilde{\mathbf{E}}^j\right)\right) \leq \frac{1}{4}$, yielding:

$$\mathrm{Var}\left(\hat{\alpha}^{\mathrm{i.i.d.}}\right) \leq \frac{1}{4n} = O\left(\frac{1}{n}\right).$$

The variance decomposes as:

$$\mathrm{Var}\left(\hat{\alpha}^{\mathrm{LOO}}\right) = \mathrm{Var}\left(\hat{\alpha}^{\mathrm{i.i.d.}}\right) + \mathrm{Var}\left(\Delta\right) + 2\mathrm{Cov}\left(\hat{\alpha}^{\mathrm{i.i.d.}}, \Delta\right).$$

We have already bounded the first term, $\mathrm{Var}\left(\hat{\alpha}^{\mathrm{i.i.d.}}\right)$. It remains to control the variance and covariance terms involving $\Delta$.

For $\mathrm{Var}\left(\Delta\right)$, note that $|\Delta| = T_1$, and we established in the proof of Theorem 3.1 that:

$$|\Delta| = T_1 \leq S_{\max}\frac{\sqrt{\frac{\log(2/\delta)}{2n}} + \frac{2}{n}}{\mu\left(S_{\min}/S_{\max} - \frac{1}{n}\right)} \quad \text{with probability} \geq 1 - \delta.$$

Thus, we obtain a tail bound:

$$\mathbb{P}(|\Delta| \ge t) \le 2 \exp\left(-2n \left(\frac{t\mu}{S_{\max}}\left(\frac{S_{\min}}{S_{\max}} - \frac{1}{n}\right) - \frac{2}{n}\right)^2\right).$$

Then:

$$\begin{aligned}
\mathrm{Var}\,(\Delta) &= \mathbb{E}\left[\Delta^2\right] - \mathbb{E}\left[\Delta\right]^2 \\
&\le \mathbb{E}\left[\Delta^2\right] \\
&= \int_0^\infty \mathbb{P}\left(|\Delta| \ge \sqrt{u}\right)\,du \\
&= 2 \int_0^\infty t\mathbb{P}\left(|\Delta| \ge t\right)\,dt \quad \left(\text{substituting } u = t^2,\ du = 2dt\right) \\
&\le 4 \int_0^\infty t \exp\left(-2n\left(\frac{t\mu}{S_{\max}}\left(\frac{S_{\min}}{S_{\max}} - \frac{1}{n}\right) - \frac{2}{n}\right)^2\right)\,dt.
\end{aligned}$$

Substituting:

$$v = \frac{t\mu}{S_{\max}}\left(\frac{S_{\min}}{S_{\max}} - \frac{1}{n}\right) - \frac{2}{n},\ dv = \frac{\mu}{S_{\max}}\left(\frac{S_{\min}}{S_{\max}} - \frac{1}{n}\right)dt,$$

we obtain:

$$\le \left(\frac{2}{\frac{\mu}{S_{\max}}\left(\frac{S_{\min}}{S_{\max}} - \frac{1}{n}\right)}\right)^2 \int_{-\frac{2}{n}}^\infty \left(v + \frac{2}{n}\right)\exp(-2nv^2)\,dv.$$

Now, we split the integral:

$$\begin{aligned}
\int_{-\frac{2}{n}}^\infty \left(v + \frac{2}{n}\right)\exp(-2nv^2)\,dv &= \int_{-\frac{2}{n}}^\infty v \exp(-2nv^2)\,dv + \int_{-\frac{2}{n}}^\infty \frac{2}{n}\exp(-2nv^2)\,dv \\
&\le \int_{-\frac{2}{n}}^\infty v \exp(-2nv^2)\,dv + \int_{-\infty}^\infty \frac{2}{n}\exp(-2nv^2)\,dv \\
&= \left[-\frac{1}{4n}\exp(-2nv^2)\right]_{-\frac{2}{n}}^\infty + \frac{2}{n}\sqrt{\frac{\pi}{2n}} \\
&= \frac{1}{4n}e^{-8/n} + \frac{2}{n}\sqrt{\frac{\pi}{2n}}.
\end{aligned}$$

Therefore:

$$\mathrm{Var}\,(\Delta) \le \left(\frac{2}{\frac{\mu}{S_{\max}}\left(\frac{S_{\min}}{S_{\max}} - \frac{1}{n}\right)}\right)^2 \left(\frac{1}{4n}e^{-8/n} + \frac{2}{n}\sqrt{\frac{\pi}{2n}}\right) = O\left(\frac{1}{n}\right).$$

Finally, by the Cauchy-Schwarz inequality:

$$\left|\mathrm{Cov}\left(\hat{\alpha}^{\mathrm{i.i.d.}}, \Delta\right)\right| \le \sqrt{\mathrm{Var}\left(\hat{\alpha}^{\mathrm{i.i.d.}}\right)\mathrm{Var}\,(\Delta)} \le \left(\frac{2}{\frac{\mu}{S_{\max}}\left(\frac{S_{\min}}{S_{\max}} - \frac{1}{n}\right)}\right)\sqrt{\frac{1}{4n}\left(\frac{1}{4n}e^{-8/n} + \frac{2}{n}\sqrt{\frac{\pi}{2n}}\right)} = O\left(\frac{1}{n}\right).$$

Putting everything together, we have shown that:

$$\mathrm{Var}\left(\hat{\alpha}^{\mathrm{LOO}}\right) = O\left(\frac{1}{n}\right).$$

$\square$

## C.2 General case

**Theorem C.6** (Theorem 3.5). *Assume that the score function $S$ is bounded and takes values in the interval $[S_{\min}, S_{\max}]$, with $0 < S_{\min} \le S_{\max}$. Suppose that the number of calibration samples satisfies $n > S_{\max}/S_{\min}$. In addition, assume that the size constraint rule and the score function satisfy the following stability property almost surely:*

*(P3) There exists a constant $L > 0$ such that:*

$$\left| \sum_{j=1}^{n} (\tilde{\alpha}_j - \mathbb{E}[\tilde{\alpha}]) \right| \leq L \max_{y \in \mathcal{Y}} \left| \sum_{j=1}^{n} \left( \mathbf{E}_y^j - \mathbb{E}\left[ \mathbf{E}_y^{\text{test}} \right] \right) \right|.$$

*Then, if the samples $(X_i, Y_i)$ are i.i.d., the leave-one-out estimator satisfies:*

$$\left| \hat{\alpha}^{\text{LOO}} - \mathbb{E}[\tilde{\alpha}] \right| = O_P \left( \frac{1}{\sqrt{n}} \right).$$

*Proof.* We have:

$$\left| \hat{\alpha}^{\text{LOO}} - \mathbb{E}[\tilde{\alpha}] \right| = \frac{1}{n} \left| \sum_{j=1}^{n} (\tilde{\alpha}_j - \mathbb{E}[\tilde{\alpha}]) \right|$$

$$\leq \frac{L}{n} \left\| \sum_{j=1}^{n} \left( \mathbf{E}^j - \mathbb{E}\left[ \mathbf{E}^{\text{test}} \right] \right) \right\|$$

$$= \frac{L}{n} \left\| \sum_{j=1}^{n} \left( \mathbf{E}^j - \tilde{\mathbf{E}}^j \right) + \left( \tilde{\mathbf{E}}^j - \mathbb{E}\left[ \tilde{\mathbf{E}}^{\text{test}} \right] \right) + \left( \mathbb{E}\left[ \tilde{\mathbf{E}}^{\text{test}} \right] - \mathbb{E}\left[ \mathbf{E}^{\text{test}} \right] \right) \right\|$$

$$\leq \frac{L}{n} \sum_{j=1}^{n} \left\| \mathbf{E}^j - \tilde{\mathbf{E}}^j \right\| + \frac{L}{n} \left\| \sum_{j=1}^{n} \left( \tilde{\mathbf{E}}^j - \mathbb{E}\left[ \tilde{\mathbf{E}}^{\text{test}} \right] \right) \right\| + L \, \mathbb{E}\left[ \left\| \tilde{\mathbf{E}}^{\text{test}} - \mathbf{E}^{\text{test}} \right\| \right],$$

where the first inequality is precisely the stability property (P3), and the second follows from the triangle inequality.

In the proof of Theorem 3.1, we have shown that:

$$\frac{1}{n} \sum_{j=1}^{n} \left\| \mathbf{E}^j - \tilde{\mathbf{E}}^j \right\| \leq S_{\max} \frac{\sqrt{\frac{\log(2/\delta)}{2n}} + \frac{2}{n}}{\mu \left( S_{\min}/S_{\max} - \frac{1}{n} \right)} \quad \text{with probability} \geq 1 - \delta,$$

and

$$\mathbb{E}\left[ \left\| \tilde{\mathbf{E}}^{\text{test}} - \mathbf{E}^{\text{test}} \right\| \right] \leq \frac{2 S_{\max}^2}{\mu S_{\min}} \frac{n+1}{n} \sqrt{\frac{\pi}{2(n+1)}}.$$

It remains to bound $\frac{1}{n} \left\| \sum_{j=1}^{n} \left( \tilde{\mathbf{E}}^j - \mathbb{E}\left[ \tilde{\mathbf{E}}^{\text{test}} \right] \right) \right\|$ with high probability. For any $t \geq 0$, union bound combined with Hoeffding's inequality yields:

$$\mathbb{P}\left( \left\| \frac{1}{n} \sum_{j=1}^{n} \left( \tilde{\mathbf{E}}^j - \mathbb{E}\left[ \tilde{\mathbf{E}}^{\text{test}} \right] \right) \right\| \geq t \right) = \mathbb{P}\left( \bigcup_{y \in \mathcal{Y}} \left| \frac{1}{n} \sum_{j=1}^{n} \left( \tilde{\mathbf{E}}_y^j - \mathbb{E}\left[ \tilde{\mathbf{E}}_y^{\text{test}} \right] \right) \right| \geq t \right)$$

$$\leq \sum_{y \in \mathcal{Y}} \underbrace{\mathbb{P}\left( \left| \frac{1}{n} \sum_{j=1}^{n} \left( \tilde{\mathbf{E}}_y^j - \mathbb{E}\left[ \tilde{\mathbf{E}}_y^{\text{test}} \right] \right) \right| \geq t \right)}_{\leq 2 \exp\left( -\frac{2 n \mu^2 t^2}{(S_{\max} - S_{\min})^2} \right)}$$

$$\leq 2 |\mathcal{Y}| \exp\left( -\frac{2 n \mu^2 t^2}{(S_{\max} - S_{\min})^2} \right).$$

Therefore:

$$\left\| \frac{1}{n} \sum_{j=1}^{n} \left( \tilde{\mathbf{E}}^j - \mathbb{E}\left[ \tilde{\mathbf{E}}^{\text{test}} \right] \right) \right\| \leq \frac{S_{\max} - S_{\min}}{\mu} \sqrt{\frac{\log(2 |\mathcal{Y}| / \delta)}{2n}} \quad \text{with probability} \geq 1 - \delta.$$

Finally, applying union bound and combining the above inequalities, we obtain:

$$|\hat{\alpha}^{\mathrm{LOO}} - \mathbb{E}[\tilde{\alpha}]| \leq LS_{\max} \frac{\sqrt{\frac{\log(4/\delta)}{2n}} + \frac{2}{n}}{\mu\left(S_{\min}/S_{\max} - \frac{1}{n}\right)} + L\frac{S_{\max} - S_{\min}}{\mu} \sqrt{\frac{\log(4|\mathcal{Y}|/\delta)}{2n}} + \frac{2LS_{\max}^2}{\mu S_{\min}} \frac{n+1}{n} \sqrt{\frac{\pi}{2(n+1)}},$$

with probability $\geq 1 - \delta$. We conclude that:

$$\left|\hat{\alpha}^{\mathrm{LOO}} - \mathbb{E}[\tilde{\alpha}]\right| = O_P\left(\frac{1}{\sqrt{n}}\right).$$

$\square$

## D   Proof of Proposition 2.2

For completeness, we give here a proof of Proposition 2.2, which can be found in [Koning, 2025a]:

**Proposition D.1.** *Consider a calibration set $\{(X_i, Y_i)\}_{i=1}^n$ and a test data point $(X_{\mathrm{test}}, Y_{\mathrm{test}})$ such that $(X_1, Y_1), \ldots, (X_n, Y_n), (X_{\mathrm{test}}, Y_{\mathrm{test}})$ are exchangeable. Let $\tilde{\alpha} > 0$ be any miscoverage level that may depend on this data. Then, for the conformal set*

$$\hat{C}_n^{\tilde{\alpha}}(x) := \left\{ y : \frac{S(x,y)}{\frac{1}{n+1}\left(\sum_{i=1}^n S(X_i, Y_i) + S(x,y)\right)} < 1/\tilde{\alpha} \right\},$$

*the coverage inequality* (2) *holds, i.e.,*

$$\mathbb{P}(Y_{\mathrm{test}} \in \hat{C}_n^{\tilde{\alpha}}(X_{\mathrm{test}})) \geq 1 - \mathbb{E}[\tilde{\alpha}].$$

*Proof.* Since $E^{\mathrm{test}}$ is an e-value (that does not depend on $\alpha$), we have $\mathbb{E}[\sup_{\alpha>0} \mathbb{1}\{E^{\mathrm{test}} \geq 1/\alpha\}/\alpha] \leq 1$, because $\mathbb{1}\{E^{\mathrm{test}} \geq 1/\alpha\} = \mathbb{1}\{\alpha E^{\mathrm{test}} \geq 1\} \leq \alpha E^{\mathrm{test}}$. Rewriting this inequality gives $1/\mathbb{E}[\sup_{\alpha>0} \mathbb{1}\{E^{\mathrm{test}} \geq 1/\alpha\}/\alpha] \geq 1$, and applying Jensen's inequality then yields $\mathbb{E}[1/(\sup_{\alpha>0} \mathbb{1}\{E^{\mathrm{test}} \geq 1/\alpha\}/\alpha)] \geq 1$, which is equivalent to $\mathbb{E}[\inf_{\alpha:E^{\mathrm{test}}\geq 1/\alpha} \alpha] \geq 1$, or, equivalently, $\mathbb{E}[1 - \inf_{\alpha:E^{\mathrm{test}}\geq 1/\alpha} \alpha] \leq 0$.

Noting that $1 - \inf_{\alpha:E^{\mathrm{test}}\geq 1/\alpha} \alpha = \sup_{\alpha:E^{\mathrm{test}}\geq 1/\alpha} \mathbb{1}\{E^{\mathrm{test}} \geq 1/\alpha\} - \alpha = \sup_{\alpha>0} \mathbb{1}\{E^{\mathrm{test}} \geq 1/\alpha\} - \alpha$, we obtain $\mathbb{E}[\sup_{\alpha>0} \mathbb{1}\{E^{\mathrm{test}} \geq 1/\alpha\} - \alpha] \leq 0$, i.e., $\sup_{\tilde{\alpha}>0} \mathbb{E}[\mathbb{1}\{E^{\mathrm{test}} \geq 1/\tilde{\alpha}\} - \tilde{\alpha}] \leq 0$.

Therefore, for any possibly data-dependent $\tilde{\alpha} > 0$, $\mathbb{E}[\mathbb{1}\{E^{\mathrm{test}} \geq 1/\tilde{\alpha}\}] \leq \mathbb{E}[\tilde{\alpha}]$, i.e., $\mathbb{P}(E^{\mathrm{test}} \geq 1/\tilde{\alpha}) \leq \mathbb{E}[\tilde{\alpha}]$. The result then follows by substituting $E^{\mathrm{test}}$ and $\hat{C}_n^{\tilde{\alpha}}$ with their definitions. $\square$

