# OpenReview forum: "Backward Conformal Prediction"
_NeurIPS.cc/2025/Conference — NeurIPS 2025 poster_

### Official Review · Reviewer_A3K2 · 2025-06-10

**Clarity:** 2
**Significance:** 1
**Originality:** 2
**Rating:** 4
**Confidence:** 4

**Summary:**

This paper studies backward conformal prediction, where we constrain the size of the prediction set while adapting the coverage level accordingly. Based on the results of Gauthier et al., this paper proposes an estimation of the miscoverage expectation so that we can estimate the expected coverage of backward conformal prediction methods. Experiments show that the estimation is pretty good in practice.

**Questions:**

Please refer to the weaknesses.

**Ethical Concerns:**

["NO or VERY MINOR ethics concerns only"]

**Final Justification:**

After the discussion, I was convinced that BCP is worth studying, so one of my main issues was addressed. Another main issue is that the contribution of this paper is not so much because it just provides an estimator of $E[\tilde{\alpha}]$. However, I believe such a contribution deserves a positive score since it makes the BCP algorithms potentially applicable in practice. So I changed my mind and decided to give Borderline Accept.

**Limitations:**

Yes.

**Paper Formatting Concerns:**

No.

**Quality:**

2

**Strengths And Weaknesses:**

### Strengths

- The paper proposes an estimation of $\mathbb{E}[\tilde{\alpha}]$ and then provides theoretical guarantees about the estimation error.


### Weaknesses

- The contribution of this paper is to provide an estimator for $\mathbb{E}[\tilde{\alpha}]$. However, the main results for backward conformal prediction are provided by Gauthier et al. All the paper does is provide an estimator, which is sadly not so meaningful due to the unknown approximation error caused by the first-order Taylor approximation in lines 134-135.
- There may be a mismatch between the claim "an estimator of $\mathbb{E}[\tilde{\alpha}]$ can help us reject predictions with low coverage" and the experimental results. The experiments show that the estimated value $\hat{\alpha}^\text{LOO}$ is close to the true value $\mathbb{E}[\tilde{\alpha}]$. However, this does not mean that the theoretical bounds here are tight. Note that given the theoretical results, we are going to trust prediction sets with $1-\hat{\alpha}^\text{LOO}- R_ \delta(n) \ge \tau$, which is a **distribution-free result**. So the experimental results are not sufficient to show the effectiveness of Backward Conformal Prediction as claimed in lines 244-245.
- In conclusion, compared with conformal prediction (CP), I do not see the necessity of studying backward conformal prediction (BCP).  Since the approximation error of the first-order Taylor approximation in lines 134-135 is not controlled, BCP can only control the prediction set size, but not the coverage level. So, it seems that CP and BCP focus on controlling different aspects of the prediction set, i.e., the coverage level and the size, respectively. From this point, it seems that BCP is still worth studying. However, the claim in this paper that we can control the size while rejecting predictions with bad coverage is not as good as controlling the coverage while rejecting predictions with large size. It is because although the former can control the size, its rules on the coverage are ambiguous due to the unknown approximation error of the Taylor approximation. On the contrary, the latter can control the coverage, and the rules for prediction set size are really easy to set. So, it seems that BCP can be regarded as a dual method for CP only if the approximation error can be controlled. By the way, in situations where errors can cause serious consequences, controlling the accuracy (coverage) is obviously more important than controlling the set size. Furthermore, the spirit of confidence sets prediction (which includes conformal prediction) is to control the coverage level but not the set size, just like the spirit of hypothesis testing is to control the type-1 error but not the type-2 error.

---

> ### Author Rebuttal · Authors · 2025-07-29
>
> *First, we would like to thank the reviewer for their time. We are confident in our approach and hope that our response will clarify our aims, the interpretation of our results, and comparison to classical statistical methodology.*
>
> * The contribution of this paper is to provide an estimator for E[α]. However, the main results for backward conformal prediction are provided by Gauthier et al. All the paper does is provide an estimator, which is sadly not so meaningful due to the unknown approximation error caused by the first-order Taylor approximation in lines 134-135.
>
> Our method indeed builds on the use of e-values in conformal prediction, in particular drawing from the work of Gauthier et al. However, that paper did not focus (at all) on obtaining marginal guarantees. In contrast, this is the central contribution of our work. We emphasize that our approach to providing the marginal guarantees (specifically the expected coverage E[α]) is entirely novel. This contribution is crucial in practice, as it enables practitioners to understand the actual coverage they can achieve with this method and make informed decisions accordingly.
>
> Regarding the final part of the comment, the Taylor expansion is never used to approximate any quantity; it is solely a technical tool to establish the coverage property: Pr(Y_test \in C(X_test)) \ge 1 - E[α]. Both the experiments from Gauthier et al. and our own results support and justify the validity of this approximation. We find empirically in a range of experiments that this coverage inequality always holds in practice.
>
> * The experiments show that the estimated value α^LOO is close to the true value E[α]. However, this does not mean that the theoretical bounds here are tight. Note that given the theoretical results, we are going to trust prediction sets with 1-α^LOO-R(n) \ge \tau, which is a distribution-free result. So the experimental results are not sufficient to show the effectiveness of Backward Conformal Prediction as claimed in lines 244-245.
>
> We are somewhat surprised by the comment about tightness, given that the 1/\sqrt{n} and 1/n convergence rates that we obtain for the mean and the variance, respectively, are ubiquitous in statistics. These rates arise from fundamental properties of random variables, probability theory, and the general structure of estimation problems.  While one could pursue analyses that establish lower bounds, that effort would duplicate statistical literature and be far beyond the scope of the paper.
>
> We certainly agree that our results are distribution-free, and we view this as a key strength of our method. We are puzzled as to how the reviewer could view this property negatively, especially since the experimental results align closely with the theoretical guarantees we establish.
>
> We would appreciate it if the reviewer could clarify what they intended by this comment. We would be happy to provide further explanation if it would help clarify our approach and make our contributions in the paper more understandable.
>
> * In conclusion, compared with conformal prediction (CP), I do not see the necessity of studying backward conformal prediction (BCP)
>
> The CP and BCP methods are fundamentally different and are designed to address two distinct problems. BCP is not intended to replace CP, but rather to be used in situations where the practitioner prioritizes control over the size of the prediction sets.
>
> * Since the approximation error of the first-order Taylor approximation in lines 134-135 is not controlled, BCP can only control the prediction set size, but not the coverage level.
>
> We do not claim that BCP controls the coverage level. Once again, the Taylor approximation is only used to establish a coverage inequality, and it is consistently satisfied in practice.
>
> Our method, however, does allow for the estimation of coverage. In a sense, BCP even provides more flexibility than CP: while CP controls the coverage but offers no way to estimate the size of the prediction sets, BCP explicitly controls the size of the prediction sets and allows for coverage estimation.
>
> * It is because although BCP can control the size, its rules on the coverage are ambiguous due to the unknown approximation error of the Taylor approximation
>
> **Again, the Taylor approximation is only used to derive the coverage inequality**; moreover, we observe that this inequality always holds in practice, which is consistent with previous results from Gauthier et al. Furthermore, our estimator allows for the estimation of the coverage guarantees.
>
> * By the way, in situations where errors can cause serious consequences, controlling the accuracy (coverage) is obviously more important than controlling the set size. Furthermore, the spirit of confidence sets prediction (which includes conformal prediction) is to control the coverage level but not the set size, just like the spirit of hypothesis testing is to control the type-1 error but not the type-2 error.
>
> The reviewer is referring to classical statistical practice, dating back to Neyman and Pearson and to Fisher.  But just because it is classical does not mean that it is always the preferred approach.  Indeed, in the paper, we provided motivating examples, for example in medicine, where controlling the size of prediction sets is crucial. In such examples, the key quantity to control is the time devoted to having experts chase down possible leads that arise from a statistical analysis.  Obtaining a large uncertainty set can be unrealistic and indeed useless if requires too much time for experts to perform a follow-up verification to whittle down the possibilities. It may be preferable to allocate expert time/effort in advance and to proceed if the estimated coverage is sufficiently large.  That’s what our method makes possible.
>
> Our method is not meant to replace existing classical approaches where the coverage level is fixed in advance. Rather, it is complementary and orthogonal, offering a new perspective on uncertainty quantification.
>
> One may view it as surprising that the statistical literature has focused entirely, for several decades, on controlling coverage, with set size a secondary consideration.  It is surprising but true.  But perhaps it is because the shift to the inverted logic of our paper is made possible by recent developments in the use of e-values and their associated post-hoc guarantees.  This makes it possible to obtain theoretical guarantees even when the coverage level is data-dependent, something that is not possible with p-values.
>
> To say this another way, classical statistical methods have been built around a focus on the control of type I error, typically with a pre-specified, data-independent significance level α and/or p-values.  Generations of statisticians and applied researchers (particularly in medicine) have criticized this focus.  Accordingly, the controversy around p-values persists.  Our work provides one way to escape the tyranny of p-values and to solve inference problems in a new way that is complementary to classical methodology.
>
> *We would like to thank the reviewer for giving us the opportunity to clarify the nature of our method relative to classical statistical practice.  Our revision will make this comparison more clear and emphasize that our approach is novel, for interesting reasons.*

---

> > ### Comment · Reviewer_A3K2 · 2025-08-03
> >
> > Thank you for your reply.
> >
> > - The contribution of this paper is to provide an estimator for E[α]. However, the main results for backward conformal prediction are provided by Gauthier et al. All the paper does is provide an estimator, which is sadly not so meaningful due to the unknown approximation error caused by the first-order Taylor approximation in lines 134-135.
> >
> > For the rebuttal about the above point. My point is that since the error of Taylor approximation is not controlled, we can not guarantee coverage $\ge 1 - E[\tilde{\alpha}]$. It means that BCP can only control the size, but not the coverage (because of uncontrolled Taylor approximation error). However, in the examples of this paper (lines 77-96), it seems that the authors __still care about the coverage__.  __In the situations where we do not care about coverage but only care about the size, the estimation of $E[\tilde{\alpha}]$  is meaningless, so the paper is meaningless.__ So, to make both BCP and the work of this paper meaningful, __we should consider both size and coverage__. Now, __based on the object that we want to make sure the size is controlled, and reject predictions with small coverage__. CP can easily do this by rejecting all predictions with a large size (the size of CP is very easy to obtain); however, for BCP, whether to reject a prediction set depends on the estimation of $E[\tilde{\alpha}]$ and the approximation error of the uncontrolled Taylor approximation. In such a comparison, I do not see much of the necessity of BCP.
> >
> > Regarding the fact that the Taylor approximation error is caused by the work of Gauthier et al but not this paper. The approximation error does exist, no matter which part causes it. So __it is not a reason to ignore the approximation error when considering the meaning of BCP__.
> >
> > - The experiments show that the estimated value α^LOO is close to the true value E[α]. However, this does not mean that the theoretical bounds here are tight. Note that given the theoretical results, we are going to trust prediction sets with 1-α^LOO-R(n) \ge \tau, which is a distribution-free result. So the experimental results are not sufficient to show the effectiveness of Backward Conformal Prediction as claimed in lines 244-245.
> >
> > Regarding the rebuttal about the above point. You do not need to be puzzled about viewing the distributional-free property negatively, since I never do that. My highlight is that current experiments are not sufficient to validate the distribution-free results. I never say distribution-free results are bad. My point is that conducting experiments on more datasets may make the paper better.

---

> > > ### Author Response · Authors · 2025-08-04
> > >
> > > *We sincerely thank the reviewer for their response and for their active participation in the rebuttal process. We will address each of the reviewer’s comments in order, with the hope of further clarifying our work.*
> > >
> > > * My point is that since the error of Taylor approximation is not controlled, we can not guarantee coverage $\ge 1 - E[\tilde{\alpha}]$
> > >
> > > We wish to note that this kind of Taylor approximation is used throughout statistics, where it is called the delta method, and it is used throughout physics and many areas of applied mathematics. While in some cases worst-case errors are controlled theoretically, such control involves assumptions, and in many cases it is considered valuable to test the approximations empirically across a wide range of conditions.  That is precisely what we have done.  We have found that our approximation is consistently satisfied in practice. All the experiments presented in our paper, as well as those by Gauthier et al., provide empirical justification, demonstrate that the inequality holds, and offer additional evidence supporting the validity of this coverage inequality.  The reviewer may come from a different community where Taylor approximation (and more generally saddlepoint approximations and asymptotic expansions) are not considered valuable, but we hope that the reviewer
> > > will recognize that there are different perspectives on this.
> > >
> > > * However, in the examples of this paper (lines 77-96), it seems that the authors still care about the coverage.
> > >
> > > Exactly, we care about both the size and the coverage. Classical statistics cares about both as well, but focuses mostly on coverage, with size as a random variable.  BCP inverts this perspective, focusing on size and allowing coverage to be random, but nonetheless providing a marginal coverage guarantee.
> > >
> > > * In the situations where we do not care about coverage but only care about the size, the estimation of $E[\tilde{\alpha}]$ is meaningless, so the paper is meaningless.
> > >
> > > We have never suggested that we care only about size and do not care about coverage. We would not suggest that as statistical practice. We kindly encourage the reviewer to revisit the paper, as this point may stem from a misunderstanding.
> > >
> > > * So, to make both BCP and the work of this paper meaningful, we should consider both size and coverage.
> > >
> > > This is exactly what we do in the paper: BCP allows us to control the size while providing marginal coverage guarantees, which can be estimated using the calibration set.
> > >
> > > * CP can easily do this by rejecting all predictions with a large size (the size of CP is very easy to obtain); however, for BCP, whether to reject a prediction set depends on the estimation of $E[\tilde{\alpha}]$ and the approximation error of the uncontrolled Taylor approximation. In such a comparison, I do not see much of the necessity of BCP.
> > >
> > > Fundamentally, conformal prediction methods are not primarily designed to accept or reject prediction sets, but rather to provide marginal guarantees. For example, in classical conformal prediction, the inferential logic is not to simply observe the size of the conformal sets and reject those that are too large. Instead, the aim is to consistently produce the smallest possible sets, a topic extensively studied in the literature. But while CP controls coverage, it does not provide any means to estimate the size of the prediction sets. Our method, in a sense, offers stronger guarantees by explicitly controlling the size of the prediction sets while still maintaining coverage guarantees. Furthermore, we emphasize again that BCP is not intended to replace CP but to complement it in scenarios where the size of the prediction sets is particularly important.
> > >
> > > * Regarding the fact that the Taylor approximation error is caused by the work of Gauthier et al but not this paper. The approximation error does exist, no matter which part causes it. So it is not a reason to ignore the approximation error when considering the meaning of BCP.
> > >
> > > We have of course not ignored the fact that we've used a Taylor approximation.  We've been very explicit about it. Note again that it is simply employed to establish the coverage inequality. Moreover, we studied the approximation in an extensive set of experiments, finding that this inequality is consistently satisfied in practice.
> > >
> > > * Regarding the rebuttal about the above point [...].
> > >
> > > We thank the reviewer for clarifying this point, and we now understand their perspective. Our experiments were initially conducted on a binary classification task, followed by the popular CIFAR-10 dataset. In ongoing work we are engaging in applications to real-world datasets, focusing on problems where controlling the size of prediction sets is particularly important.
> > >
> > > *We thank the reviewer once again for their time and hope that our method and its scope are now clearer. We appreciate the opportunity to provide further clarification.*

---

> > > > ### Author Response · Authors · 2025-08-05
> > > >
> > > > Let us add that we really do think that size control is primary in some domains. Returning to our example of the doctor, if they ask for coverage of 95%, they may obtain a large set of possible diseases or a small set. Knowing that the true disease is in the set (whp) is not super helpful. The doctor only has the time and resources to investigate (e.g., with follow-up tests) a few possibiiities. Our approach would allow him/her to (say) ask for four candidates, which may be what he/she can afford to follow up on. If that was the end of the story, it wouldn't be satisfying. But our approach does give an estimate of coverage (confidence) to the doctor. He/she can look at that (random) confidence and decide if they're OK with it. If it's a large number, near one, they'll probably proceed. If it's a small number, they will do what a good honest statistician always does---which is to get additional data, perhaps returning to the patient and explaining the situation.

---

> > > > > ### Comment · Reviewer_A3K2 · 2025-08-05
> > > > >
> > > > > Thank you for the further discussion. Based on your example of the doctor, let us make it clear why I think BCP is not as preferred as CP (in my opinion).
> > > > >
> > > > > In this case, if we use CP with a predefined confidence level (e.g. $0.95$), then for each patient $x$, the doctor will get a prediction set $C(x)$. If the doctor wishes the set size $\le 4$, then he just needs to consider the size $|C(x)|$, if $|C(x)| > 4$, then the doctor can return to the patient and explain the situation.
> > > > >
> > > > > As a comparison:
> > > > > - For CP, coverage is guaranteed, and the set size $|C(x)|$ is easy to get. They are strictly satisfied (coverage) and accurate (size).
> > > > > - For BCP, the set size $|C(x)|$ is strictly satisfied. However, the coverage is estimated, and there are two errors in the estimation. The first is the error of estimating $E[\tilde{\alpha}]$, which can be controlled. The second is the Taylor approximation error, which is not controlled as far as the paper shows.
> > > > >
> > > > > So, why do we choose a constrained set size and estimated coverage over the constrained coverage and accurate size? It seems that controlled + estimated $<$ controlled + accurate.
> > > > >
> > > > > In fact, my concerns about the paper lie in two main points. One of them is the necessity of BCP, as mentioned above. The other is that I think the paper is an incremental work of Gauthier et al, which only proposes an estimator, and I think the contribution is not enough as a NeurIPS paper.
> > > > >
> > > > > If the authors can convince me about the two points, I am willing to increase my score accordingly.

---

> > > > > > ### Author Response · Authors · 2025-08-05
> > > > > >
> > > > > > *We sincerely thank the reviewer for their active participation in the rebuttal process and their engagement, as well as for their thoughtful question that helps highlight the contributions of our work.*
> > > > > >
> > > > > > Regarding the doctor's example and the usefulness of BCP, we would like to recall that **conformal prediction provides marginal guarantees, and that our goal is precisely to obtain marginal guarantees as well**. We believe the reviewer's comparison is not entirely fair: it is true that in standard conformal prediction (CP), for a given sample, one obtains a controlled coverage level and a conformal set whose size can be observed. Similarly, in BCP, we obtain a prediction set with controlled size, **along with an adaptive coverage level** $\tilde{\alpha}$ (rather than no coverage information at all or simply an estimation or marginal coverage, as suggested).
> > > > > >
> > > > > > However, it is important to note that **in standard CP, there are no marginal guarantees on the average size of the conformal sets obtained**. In contrast, **in BCP, we provide additional guarantees on the average coverage achieved across samples.**
> > > > > >
> > > > > > These guarantees are based on two steps:
> > > > > >
> > > > > > 1) A Taylor approximation of the expected size ratio $\mathbb{P}(Y_{\rm test} \not \in C(X_{\rm test}) | \tilde{\alpha}) / \tilde{\alpha}$, which yields a coverage inequality. And this inequality is consistently satisfied in practice.
> > > > > >
> > > > > > 2) An estimation of $\mathbb{E}[\tilde{\alpha}]$ based on the calibration set.
> > > > > >
> > > > > > At no point are these two steps approximating the same quantity in a way that would lead to an accumulation of errors (with one of the two being uncontrolled). The first is used solely to establish a coverage bound (which is empirically reliable). It is not an uncontrolled approximation.
> > > > > >
> > > > > > Therefore, BCP provides more guarantees, in a purely accounting sense.
> > > > > >
> > > > > > But beyond that, BCP also provides post-hoc guarantees that allow practitioners to navigate the Pareto frontier between coverage and average prediction set size (see our response to reviewer i8Nu), which we believe constitutes a statistically deeper contribution. Standard statistical methods only allow selecting a single point on the Pareto frontier (by fixing coverage in advance), whereas our reverse approach enables choosing $\alpha$ in a data-dependent manner, thus allowing navigation along the entire Pareto curve.
> > > > > >
> > > > > > Moremove, from a methodological perspective, BCP complements CP by shifting the focus toward prediction set size : an aspect that can be crucial in practice, particularly in fields like medicine where the number of hypotheses considered must often be strictly limited.
> > > > > >
> > > > > > Regarding the contribution itself, we believe that having marginal guarantees on $\mathbb{E}[\tilde{\alpha}]$ without being able to estimate it does not provide the practitioner with any actionable information. In practice, the estimation of this quantity is crucial. Without it, the practitioner gains no new insight. And ultimately, that is what statistics is about: obtaining information that enables better and more informed decisions.
> > > > > >
> > > > > > Our work operationalizes the idea hinted at in Gauthier et al., making it practical and directly usable. Furthermore, the theoretical analysis is both technically novel and statistically meaningful, opening new perspectives in the study of conformal methods. While we deliberately chose to keep the main exposition focused on methodology, the results are supported by substantial mathematical work, which is presented in detail in the appendix.
> > > > > >
> > > > > > To the best of our knowledge, our paper is the first to leverage the calibration set not only for running the conformal prediction algorithm, but also as a source of additional information, enabling post-hoc statistical guarantees that extend the utility of conformal prediction in practice.
> > > > > >
> > > > > > In conclusion: our work proposes an estimation of guarantees without which the method would be ineffective, thus addressing a crucial problem. It also provides a technically novel mathematical proof that could potentially be applied to other statistical problems. Furthermore, it develops innovative ideas, such as leveraging the calibration set to obtain additional guarantees. Finally, the theoretical results are supported by a series of experiments that validate and illustrate the practical relevance of our approach.

---

> > > > > > > ### Comment · Reviewer_A3K2 · 2025-08-05
> > > > > > >
> > > > > > > Thank you for your reply.
> > > > > > >
> > > > > > > In your reply, you said that BCP provides more guarantees than CP. Let us just ignore the estimation error of the coverage in BCP. Perhaps we have different ways to understand the word "guarantee". To make it clearer, in the following, I use the term **control $a$** to stand for the fact that we can make $a \le$ some value or $a \ge $ some value we predefined, and **calculate $a$** to stand for the fact that we can know the value of $a$. It seems that BCP can control set size and calculate the coverage, while CP can control the coverage and calculate the set size. In this sense, it seems that CP and BCP both have one guarantee (control). Now, considering the ability to calculate, CP does not need estimation, while BCP needs.
> > > > > > >
> > > > > > > Now, considering the example of the doctor, since we care both coverage and size, assume we want the prediction set $C$ to satisfy $|C(x)| \le \mathcal{T}(x)$ (fix the calibration set) and $P(y \in C(x)) \ge 1-\alpha$. If $C(x)$ can not satisfy both requirements, the doctor will return to the patient and explain the situation (we call this a rejection). Then CP can reject $C(x)$ with $|C(x)| > \mathcal{T}(x)$ and BCP can reject $C(x)$ with estimated coverage $< 1 - \alpha$. I think that BCP is not worth studying because, according to the goal $|C(x)| \le \mathcal{T}(x)$ and $P(y \in C(x)) \ge 1-\alpha$ (which is a reflection that we care about both coverage and size), although CP and BCP can both achieve this goal, BCP relies on an estimation and Taylor approximation. **This makes CP more preferable than BCP when we care about both coverage and size.** I know the authors said that the approximation works well in practice; however, if we have an approach that does not need approximation, why do we take the risk of using a method with approximation whose approximation error is not controlled (here, not controlled means that the approximation error is not considered to adjust the set $C$ to make coverage $\ge 1 - E[\tilde{\alpha}]$ hold strictly)? Moreover, I do not understand the claim "We believe the reviewer's comparison is not entirely fair." Could you please explain it in detail?
> > > > > > >
> > > > > > > Now I read the answer about Pareto frontier, it seems that given $\alpha$, CP can not calculate the average size while given $\tilde{\alpha}$, BCP can estimate the coverage. This means that BCP has the potential to navigate the Pareto frontier (although the estimation is required) while CP does not. I appreciate it and agree that there seem to be some reasons to study BCP. I am willing to increase my score accordingly. However, could you please answer my questions in the last paragraph? Different from the Pareto frontier view, it is a different view of comparing CP and BCP.

---

> > > > ### Comment · Reviewer_A3K2 · 2025-08-05
> > > >
> > > > Thank you for your reply. I am curious why the authors like to separate my review and understand it separately.
> > > >
> > > > In my original review, I said that since the theoretical results are distribution-free, the experiments on CIFAR-10 are not enough. However, the authors cut my review into two pieces and accused me of regarding distribution-free results negatively. It is the first time, and I did not pay much attention to it.
> > > >
> > > > In my follow-up review, my full sentence is: "**In the situations where we do not care about coverage but only care about the size, the estimation of $E[\tilde{\alpha}]$ is meaningless, so the paper is meaningless. So, to make both BCP and the work of this paper meaningful, we should consider both size and coverage**". My final point is that to evaluate the paper, we should consider both size and coverage. However, the authors cut my review into two pieces. The first piece is "In the situations where we do not care about coverage but only care about the size, the estimation of $E[\tilde{\alpha}]$ is meaningless, so the paper is meaningless". The authors think I do not read the paper carefully. It is the second time.
> > > >
> > > > I recommend that the authors read reviews carefully and understand the overall meaning of the reviews rather than refuting them separately.

---

> > > > > ### Author Response · Authors · 2025-08-05
> > > > >
> > > > > We thank the reviewer for their response. Regarding the reviewer’s initial comment about distribution-free results and experiments, we must admit that we did not fully understand the intended point at first, particularly the connection between distribution-free guarantees and the experimental setup. That is why we wrote: “We would appreciate it if the reviewer could clarify what they intended by this comment.” We thank the reviewer for doing so in their second reply, which allowed us to better grasp their perspective. Accordingly, we followed up with: “We thank the reviewer for clarifying this point, and we now understand their perspective.” In this context, we believe the statement “the authors cut my review into two pieces and accused me of regarding distribution-free results negatively” is somewhat of an overstatement. We certainly did not mean to accuse anyone; our intention was solely to understand a point that, to us, was not initially clear. We sincerely appreciate the reviewer’s effort in providing a clarification.
> > > > >
> > > > > As for the substantive content of the reviewer’s comment (in particular, the remark that our response addressed the comment in two parts) we would like to emphasize that this structure in no way weakens the overall response. Our method does not focus solely on the size of the prediction sets, which is why we believe the first part of the reviewer’s comment ("In the situations where we do not care about coverage but only care about the size, the estimation of $E[\tilde{\alpha}]$ is meaningless, so the paper is meaningless") is not entirely relevant in this context. We consider both size and coverage: our approach explicitly controls the size of the prediction sets and provides valid marginal coverage guarantees.

---

> ### Author Response · Authors · 2025-08-06
>
> *We thank the reviewer for their feedback and continued engagement in the rebuttal process, as well as for their thoughtful comment that helps clarify the decision-making benefits of our method. We have enjoyed the interaction.*
>
> First, we are pleased to see that the reviewer recognizes the statistical strengths of our approach, particularly from the perspective of the Pareto frontier. We have also added a comment on this topic in our response to reviewer i8Nu, and we invite the reviewer to refer to it for further details.
>
> The reviewer also raises an important point regarding the accounting aspect of the method, suggesting that CP might be more appropriate in this regard since it does not rely on approximations. We will address the reviewer’s broader concern holistically, rather than dissecting it point by point, in line with the reviewer’s stated preference. We believe that the comparison made by the reviewer is somewhat unfair and inaccurate, even from an accounting perspective, as BCP provides stronger guarantees. Let us explain.
>
> First, we recall that conformal prediction provides **marginal** guarantees. That is, for any given point, conformal prediction yields a set with population-level reliability, not individual certainty. So for a new patient, neither CP nor BCP can guarantee coverage on that individual case. Both methods offer guarantees of the form: "I know that over the long run, this method will include the true label at least $1-\alpha$ (or $1-\mathbb{E}[\tilde{\alpha}]$ in the case of BCP) fraction of the time, but I don’t know anything specific about this particular case."
> This limitation of conformal methods has been the subject of growing attention in the literature, which now explores conditional guarantees (see, for example [1]). But in their standard form, conformal methods provide only marginal guarantees.
>
> For this reason, we believe it is not appropriate to evaluate the method based on a single example. It is more appropriate to assess the behavior from a marginal, population-level perspective. Accordingly, we do not frame “rejection” at the level of individual predictions. Instead, in our paper, rejection refers to rejecting an algorithm configuration as a whole. For instance, if a size constraint rule is too restrictive and leads to a predicted coverage ($1-\alpha^{\rm LOO} \approx 1-\mathbb{E}[\tilde{\alpha}]$) that is too low, then the practitioner may reject that algorithm because the marginal guarantees are insufficient for their needs. They may then choose a less restrictive size constraint rule to achieve higher average coverage.
>
> In summary, decisions in conformal prediction are made at the population level, not at the level of single instances. We hope this clarifies our perspective and would be happy to add a clarifying remark in the final version of the paper if the reviewer feels it would improve the reader’s understanding.
>
> Furthermore, we believe that even from an accounting standpoint, BCP provides stronger guarantees. We summarize the benefits of each method below, using the reviewer’s helpful "control"/"calculate" terminology, and distinguishing between individual-level, validity across all samples, and marginal guarantees (which, as discussed above, are central in conformal prediction):
>
> * For CP:
>
> Coverage Guarantee: control (across all samples, corresponds to $\alpha$),
>
> Set Size Guarantee: calculate (at the individual level, corresponds to $|C(x)|$),
>
> Marginal Guarantee: none.
>
> * For BCP:
>
> Coverage Guarantee: calculate (across all samples, corresponds to $\tilde{\alpha}$),
>
> Set Size Guarantee: control (at the individual level, corresponds to $\mathcal{T}(x)$),
>
> Marginal Guarantee: estimation of average coverage (marginal, corresponds to $\mathbb{E}[\tilde{\alpha}]$)
>
> Thus, even from the perspective proposed by the reviewer, which differs from the Pareto viewpoint, BCP still provides stronger guarantees. This is why **we believe BCP to be preferable when one cares about both coverage and set size**. BCP provides more information than CP, which ultimately allows for better statistical decision-making.
>
> We also want to emphasize that we do not claim BCP is intended to replace CP, but rather to complement it, even if we believe that from both the reviewer’s perspective and the Pareto frontier perspective, BCP offers stronger and more practical guarantees. We believe BCP is particularly useful in settings where controlling set size matters, such as in medical applications.
>
> *We hope this response addresses the reviewer’s concerns, and we would be happy to clarify further if any questions remain.*
>
> References: [1] Gibbs, Isaac, John J. Cherian, and Emmanuel J. Candès. "Conformal prediction with conditional guarantees." Journal of the Royal Statistical Society Series B: Statistical Methodology (2025): qkaf008.

---

> > ### Comment · Reviewer_A3K2 · 2025-08-08
> >
> > Thank you for your reply. I am sorry for the late reply due to missing the email from openreview.
> >
> > After your explanation, I think I understand your point. It seems that CP can control the coverage, but if I reject the ones that do not satisfy the size constraint, the whole algorithm (with rejection) may not satisfy the $1-\alpha$ coverage guarantee because it is not the original CP algorithm. I admit that from this point, my comparison is not so fair.
> >
> > After the rebuttal, I am convinced and believe that BCP is a complement of CP and is worth studying. Thank you for your patient reply. I enjoy the discussion very much.
> >
> > Based on the above situation, I am willing to increase my score.

---

> > > ### Author Response · Authors · 2025-08-08
> > >
> > > We would like to thank the reviewer's thoughtful engagement with our work and the time taken to reconsider our paper. We are pleased that our previous response helped clarify the relationship between CP and BCP, and we are grateful that the reviewer now recognizes BCP's role as a meaningful complement to standard conformal prediction methods.
> > >
> > > In light of this discussion, we plan to add a brief but explicit discussion in the final version of the paper to better highlight this complementary relationship. We believe this addition will help future readers better situate BCP within the existing conformal prediction framework.
> > >
> > > We are truly grateful for the reviewer's constructive feedback throughout this process, which has helped strengthen both our presentation and the paper's overall contribution.

---

### Official Review · Reviewer_kQ7Q · 2025-06-28

**Clarity:** 3
**Significance:** 3
**Originality:** 4
**Rating:** 5
**Confidence:** 4

**Summary:**

This paper introduces Backward Conformal Prediction, a method that ensures conformal coverage while offering flexible control over prediction set sizes. Unlike standard conformal prediction, which fixes the coverage level and allows set sizes to vary, this approach defines a size constraint rule to dictate how prediction set sizes should behave based on observed data and adapts the coverage level accordingly. It builds on two key foundations: post-hoc validity using e-values to ensure marginal coverage and a novel leave-one-out estimator (α^LOO) to compute the marginal miscoverage from the calibration set, making theoretical guarantees practical. The method is particularly useful in applications like medical diagnosis where large prediction sets are impractical. Theoretical results show the estimator's consistency, and experiments on CIFAR-10 and binary classification tasks demonstrate that it maintains computable coverage guarantees while controlling prediction set sizes effectively. The approach prioritizes set size control over fixed coverage, offering a flexible framework for uncertainty quantification in machine learning.

**Questions:**

See above.

**Ethical Concerns:**

["NO or VERY MINOR ethics concerns only"]

**Final Justification:**

Post-rebuttal: The authors' replies (as well as their discussion with other reviewers) address my concerns properly. Therefore I keep my score of 5 unchanged.

**Limitations:**

See above.

**Quality:**

3

**Strengths And Weaknesses:**

(+)
1. This paper focuses on an important topic of uncertainty quantification, with a popular technique conformal prediction.
2. The idea in this paper is pretty novel. To the best of my knowledge, I did not find papers similar to this one, which starts from the constraints of the confidence interval.
3. Based on the second point, I am convinced that the constraints of the confidence interval are indeed important in many fields like medicine.
4. The authors further provide theoretical and empirical evidence to validate their findings.

(-)
1. Could the authors provide evidence on why this paper uses conformal e-prediction as the basis? Is it necessary for the backward conformal prediction? What if we directly use other conformal prediction techniques?
2. The experiments part might be a little bit weak. More realistic experiments might enhance the scope of this paper.

---

> ### Author Rebuttal · Authors · 2025-07-29
>
> *We would like to thank the reviewer for their detailed review highlighting the strengths of our paper, and we are grateful to know that they appreciated the work. We are also pleased to see that the reviewer found the paper to be well written, clear, original, and addressing an important problem. We will provide a few additional comments on the strengths and questions raised by the reviewer, which are closely aligned with those of reviewer i8Nu.*
>
> * The idea in this paper is pretty novel. To the best of my knowledge, I did not find papers similar to this one, which starts from the constraints of the confidence interval
>
> This idea is indeed novel, and we are pleased to see that the reviewer shares our enthusiasm for our new method, which quantifies uncertainty based primarily on a size constraint. It is a promising approach that we believe has potential applications beyond the scope of this paper.
>
> * I am convinced that the constraints of the confidence interval are indeed important in many fields like medicine
>
> We appreciate that the reviewer shares our view on the importance of our method, particularly in fields such as medicine, as illustrated in the paper. We note that reviewer i8Nu agrees on the usefulness of size-constrained uncertainty quantification. However, we are surprised to see that reviewer A3K2 does not share this common enthusiasm.
>
> * The authors further provide theoretical and empirical evidence to validate their findings
>
> We thank the reviewer for highlighting the theoretical contribution of this paper, as well as the experiments that validate our theoretical findings.
>
> * Could the authors provide evidence on why this paper uses conformal e-prediction as the basis? Is it necessary for the backward conformal prediction? What if we directly use other conformal prediction techniques?
>
> We thank the reviewer for their question, which provides an opportunity to highlight the importance of using conformal e-prediction in this context.
>
> First, traditional conformal prediction methods are not designed to control the size of prediction sets, which can be problematic in certain applications (medicine, for instance). We propose a method that is dual to conformal prediction, shifting the focus from controlling coverage at a fixed level to prioritizing control over the size of prediction sets. Our approach is not intended to replace conformal prediction; rather, we believe it complements it and may be more suitable in scenarios where controlling the size of prediction sets is critical.
>
> Regarding the use of e-values, their key advantage lies in the post-hoc guarantees they enable. This allows us to retain theoretical guarantees even when the size-constraint rule depends on the data. Simpler alternatives may exist in more restricted settings, as noted by reviewer i8Nu. However, our method is more general and remains valid even when the size-constraint rule is data-dependent.
>
> We would be happy to add a remark in the final version of the paper to clarify why we rely on e-values in this setting rather than on alternative methods that only apply to the simpler case.
>
> * The experiments part might be a little bit weak. More realistic experiments might enhance the scope of this paper.
>
> We demonstrated the validity of our method on a simple binary classification example and experiments on the popular CIFAR-10 dataset. In future work, we would be happy to apply our method to real-world datasets and even deploy it in practical settings, such as medical applications.
>
> *We once again thank the reviewer for their detailed review and their enthusiasm for our work.*

---

> > ### Comment · Reviewer_kQ7Q · 2025-08-09
> >
> > I thank the authors' responses and keep my positive score unchanged.

---

### Official Review · Reviewer_i8Nu · 2025-07-03

**Clarity:** 4
**Significance:** 2
**Originality:** 4
**Rating:** 4
**Confidence:** 4

**Summary:**

This paper proposes "backward conformal prediction" a conformal method that instead of fixing the coverage before-hand, it fixes the set size, and then make prediction sets that admits to that set size constraint aiming for largest possible coverage guarantees. Their method rely on recent advancements in connecting e-values to conformal prediction. Furthermore, the introduce a novel leave-one-out style estimator of the achievable coverage level of their conformal method. They also provide empirical evidence for the performance of their method.

**Questions:**

Already asked above.

**Ethical Concerns:**

["NO or VERY MINOR ethics concerns only"]

**Final Justification:**

I have read other reviewers comments, and I believe some of the points raised by the reviewer A3K2 are valid and has to be addressed properly. For instance, the point about approximation error is valid to me. Also, about the importance of set size control CP, although I find it interesting, the use of it in practice is not entirely straightforward. The thing about coverage control is that it has some clear implications for the down-stream decision makers, for instance look at the recent work of [1]. It is not obvious how a set size controlled prediction set should be used or affect the decisions. This needs more discussions.

That being said, I still believe this is a novel point of view to CP and can benefit the community potentially with opening new perspectives and research directions.

**Limitations:**

yes.

**Paper Formatting Concerns:**

no.

**Quality:**

3

**Strengths And Weaknesses:**

- The idea of providing set-size constrained CP sets is interesting and useful for a set of real-world applications.
- The presentation of the paper is very good. I like the easy-to-understand introduction to e-values and their application to cp without making the notations messy.
- The LOO method to estimate the coverage level sounds novel to me, and might also be of independent interest beyond CP.
- Despite the pleasant characteristics of e-values, I am still not fully convinced why e-values are actually needed in this scenario, at least from the algorithmic perspective. One can propose the following, perhaps simpler, algorithm: Take any conformity score you like. Look at the prediction sets of the form ${y | S(x, y) \leq q}$, as is the convention in CP. Then sweep the value of q. if you pick q small enough, you will eventually satisfy the set size constraint. Then one can pick the largest value of q such that the set size constraint is satisfied over the calibration points. Then, perhaps by some very mild assumptions (iid and bounded scores) one can derive guarantees for set size constraint at test time. Similarly, by looking at what quantile of the calibration scores the selected q belongs to, one can also derive an estimate (along with a guarantee) for the realized coverage in test-time. How is this algorithm different than yours? and why should some one take the route of (potentially more complicated) e-values?

---

> ### Author Rebuttal · Authors · 2025-07-29
>
> *First, we would like to thank the reviewer for their time and detailed review. We also appreciate all the positive feedback received regarding the presentation, clarity, and originality of the paper. We will respond to all comments in order and also address the main question raised by the reviewer regarding the benefits of using e-values for this problem.*
>
> *We note that the points made by the review are similar to that of reviewer kQ7Q, and that our response to their review may contain additional complementary explanations that could help guide the reviewer in their comments.*
>
> * The idea of providing set-size constrained CP sets is interesting and useful for a set of real-world applications
>
> We appreciate that the reviewer shares our perspective on the value of size-constrained uncertainty quantification. We believe it is indeed a key method with important real-world applications, notably in medicine, as highlighted in our paper. This view is also supported by reviewer kQ7Q. However, reviewer A3K2 does not find this aspect to be of any relevance, which we found somewhat surprising.
>
> * The LOO method to estimate the coverage level sounds novel to me, and might also be of independent interest beyond CP
>
> We are pleased that the reviewer perceives the scope of our new method even though we preferred to highlight its simplicity in our presentation. Indeed, we specifically designed a proof tailored to our problem, based on an analytical study of the quantile function combined with probabilistic inequalities. This proof could indeed be reused as is, adapted, or serve as inspiration for tackling other problems. The LOO method is indeed novel, and we hope it may inspire similar types of proofs in the literature.
>
> * Despite the pleasant characteristics of e-values, I am still not fully convinced why e-values are actually needed in this scenario, at least from the algorithmic perspective
>
> First, we understand the simple algorithm and method proposed by the reviewer. It is indeed possible to obtain theoretical guarantees under the assumption that the data are i.i.d. However, this only works in the simple case **where the size-constraint rule does not depend on the data**. While we did treat this case in the paper, it was primarily to build intuition for the proof in the more general setting **where the size-constraint rule does depend on the data (including all the calibration data), and the confidence level is random**. To achieve sound theoretical guarantees in this more general setting it’s essential to make use of the post-hoc validity that is provided by e-values.
>
> To help future readers to understand this (critical, but somewhat subtle) issue, we will make use of the reviewer’s suggested algorithm as a didactic device to compare and contrast to our general approach.  In particular, we will add a remark comparing the different approaches in the case where the size-constraint rule is constant.
>
> *We would like to thank the reviewer for their useful and insightful comments.*

---

> ### Comment · Reviewer_i8Nu · 2025-08-04
>
> I thank the authors for their detailed response. In particular, the point about data-dependent set size control makes sense to me, hence I keep my positive evaluation.
>
> That being said, I have read other reviewers comments, and I believe some of the points raised by the reviewer A3K2 are valid and has to be addressed properly. For instance, the point about approximation error is valid to me. Also, about the importance of set size control CP, although I find it interesting, the use of it in practice is not entirely straightforward. The thing about coverage control is that it has some clear implications for the down-stream decision makers, for instance look at the recent work of [1]. It is not obvious how a set size controlled prediction set should be used or affect the decisions. This needs more discussions.
>
> [1]: Decision Theoretic Foundations for Conformal Prediction: Optimal Uncertainty Quantification for Risk-Averse Agents, Kiyani et. al.

---

> > ### Author Response · Authors · 2025-08-04
> >
> > We thank the reviewer for their constructive comments. We suggest taking a look at our updated rebuttal to reviewer A3K2, where we directly address these points in more detail. We hope this helps clarify our perspective.

---

> ### Comment · Reviewer_i8Nu · 2025-08-04
>
> I have read the second response to reviewer A3K2 as well.
>
> "Exactly, we care about both the size and the coverage. Classical statistics cares about both as well, but focuses ..."
>
> This is an interesting perspective. However, it is still not exactly obvious how it is going to be useful. Perhaps it would be useful to think of a Pareto optimal curve where on one axis we have coverage and on the other we (minimum) average set size (possible). The more coverage you demand, it would NECESSARILY lead to a larger average set size. Hence, in principle, this reverse approach is just giving a different parametrization for the same Pareto curve. Which is interesting by itself to me, but the authors response to the reviewer A3K2 looks like a bit of overclaim in terms of the impact of this point of view.
>
> To see why this Pareto curves matter, one has to pay attention that, if the decision maker (e.g. the doctor) is not satisfied with the size of a normal CP set, and wants a smaller one, they would have to necessarily sacrifice some coverage. There is no magic (unless the authors are claiming it, but I dont see it). Now here it comes the question of whether it makes sense from the point of view of a decision maker to sacrifice coverage for a controlled set size. And this DOES need a thorough discussion from a decision making point of view. I am not saying it is not useful, I am just saying it is not entirely obvious, as it is stated by the authors.

---

> > ### Author Response · Authors · 2025-08-05
> >
> > Thanks for continuing to engage.  Yes, statistics has had that Pareto curve in mind since the days of Wald, who originated decision theory in the 30s.  To make it more than a metaphor, though, in the face of data (which distinguishes statistics from economics), one needs a method to find points---any points---on or near the curve.  As we noted, the Neyman-Pearson approach focuses on coverage, and size is left random, and there is surprisingly little theory on how to exert even any control on size in the NP paradigm.  One can (and people do) criticize the paradigm for this, but it's survived for a long time.  Our approach inverts the NP paradigm, making size the focus, with coverage random.  We do think that we improve on NP in that at least we say something (marginal coverage) about our secondary objective.
> >
> > Lastly, we really do think that size control is primary in some domains.  Returning to our example of the doctor, if they ask for coverage of 95%, they may obtain a large set of possible diseases or a small set.  Knowing that the true disease is in the set (whp) is not super helpful.  The doctor only has the time and resources to investigate (e.g., with follow-up tests) a few possibiiities.  Our approach would allow him/her to (say) ask for four candidates, which may be what he/she can afford to follow up on.  If that was the end of the story, it wouldn't be satisfying.  But our approach does give an estimate of coverage (confidence) to the doctor.  He/she can look at that (random) confidence and decide if they're OK with it.  If it's a large number, near one, they'll probably proceed.  If it's a small number, they will do what a good honest statistician always does---which is to get additional data, perhaps returning to the patient and explaining the situation.

---

> > > ### Author Response · Authors · 2025-08-06
> > >
> > > Let us also add a clarification regarding the Pareto curve. Suppose the size-constraint rule is chosen by the practitioner to minimize the average size of the prediction sets. In that case, we are indeed in the exact setting described by the reviewer, where we move along the classical Pareto curve that trades off coverage and set size. However, we respectfully disagree with the claim that our method merely reparametrizes this curve without added utility.
> > >
> > > A key limitation of the traditional framework is that the coverage level must be fixed in advance, independently of the test instance. This amounts to selecting a single point on the Pareto frontier (e.g., "I want 90% coverage"), with no possibility of adapting to specific test-time constraints.
> > >
> > > In contrast, our backward aproach allows the coverage level to be chosen adaptively. This enables us to move along the Pareto frontier in a data-dependent fashion and select any optimal trade-off point on the curve (a flexibility that is fundamentally out of reach in the classical setting). In fact, in the NP framework, adapting the coverage after seeing data (i.e., trying different coverage values to improve average size) is known as "p-hacking" and undermines the validity guarantees.
> > >
> > > Not only does our approach allow valid, adaptive navigation along the standard Pareto curve, it also supports greater flexibility: by choosing a different size-constraint rule (not necessarily the one minimizing average size), one can define new Pareto curves with different y-axes and navigate those instead, something the classical approach cannot accommodate.

---

### Decision · Program_Chairs · 2025-09-17

**Decision:**

Accept (poster)

**Comment:**

This paper introduces Backward Conformal Prediction (BCP), a variant of conformal prediction that inverts the usual framework: instead of fixing coverage and allowing prediction set sizes to vary, it fixes the set size in advance and then derives prediction sets that maximize coverage under this constraint. The method uses connections between e-values and conformal prediction, contributing a novel LOO estimator of achievable coverage, which the authors show to be consistent and practically useful. Empirical studies suggest that BCP can maintain reliable coverage guarantees while controlling set size. This feature is particularly appealing in settings like medical diagnosis where unwieldy prediction sets are impractical.

Reviewers have generally found the idea of set-size constrained prediction sets both interesting and relevant for real-world applications, with a clear and well-presented introduction to e-values. The LOO coverage estimator is regarded as novel and potentially useful beyond this paper. The work is also appreciated for its theoretical grounding and empirical validation, with novelty in shifting the focus of conformal prediction toward size constraints rather than coverage constraints. However, several weaknesses were noted. Some argue that the reliance on e-values is not fully justified and that simpler, more direct algorithms might achieve similar outcomes without the added complexity. Others note that the main contributions of BCP were already introduced by Gauthier et al., with this paper primarily providing an estimator whose value is diminished by uncontrolled approximation errors (e.g., the first-order Taylor expansion). The experimental section is seen as somewhat limited in scope and lacking more realistic applications. Conceptually, two reviewers also questioned whether prioritizing set size over coverage aligns with the spirit of conformal prediction, since coverage guarantees are traditionally considered more fundamental, especially in high-stakes domains.

After extensive discussion between the authors and the reviewers, I believe the authors have successfully addressed their concerns. Overall, the paper offers a promising and original direction, but it could benefit from expanding discussion around its necessity, methodological choices, and experimental validation. In particular, the authors are encouraged to include many of the points raised during the discussion period strategically in the paper.